# A User's Guide to Sampling Strategies for Sliced Optimal Transport

**Keanu SISOUK**                                                    *Keanu.Sisouk@lip6.fr*
*CNRS, LIP6*
*Sorbonne University*

**Julie Delon**                                                    *julie.delon@u-paris.fr*
*MAP5*
*Paris Cite University*

**Julien Tierny**                                          *julien.tierny@sorbonne-universite.fr*
*CNRS, LIP6*
*Sorbonne University*

**Reviewed on OpenReview:** *https://openreview.net/forum?id=ECBepTWAFG&noteId=icMjAML5rH*

## Abstract

This paper serves as a user's guide to sampling strategies for sliced optimal transport [Rabin et al., 2012; Bonneel et al., 2015]. We provide reminders and additional regularity results on the Sliced Wasserstein distance. We detail the construction methods, generation time complexity, theoretical guarantees, and conditions for each strategy. Additionally, we provide insights into their suitability for sliced optimal transport in theory. Extensive experiments on both simulated and real-world data offer a representative comparison of the strategies, culminating in practical recommendations for their best usage.

## 1 Introduction

The Wasserstein distance is acclaimed for its geometric relevance in comparing probability distributions. Having gathered a lot of theoretical work [Santambrogio, 2015; Villani, 2008], it has also proved to be relevant in numerous applied domains in the last fifteen years, such as image comparison [Rabin et al., 2009], image registration [Feydy et al., 2017], domain adaptation [Courty et al., 2016], generative modeling [Arjovsky et al., 2017; Gulrajani et al., 2017; Salimans et al., 2018], inverse problems in imaging [Hertrich et al., 2022] or topological data analysis [Edelsbrunner & Harer, 2009; Pont et al., 2021], to name just a few. The computational demands of the Wasserstein distance are, however, quite high, since evaluating the distance between two discrete distributions of $N$ samples with traditional linear programming methods incurs a runtime complexity of $O(N^3 \log N)$ [Peyré et al., 2019]. This computational burden has motivated the development of alternative metrics sharing some of the desirable properties of the Wasserstein distance but with reduced complexity.

The Sliced Wasserstein (SW) distance [Rabin et al., 2012; Bonneel et al., 2015], defined by slicing the Wasserstein distance along all possible directions on the hypersphere, is one of these efficient alternatives. Indeed, the SW distance maintains the core properties of the Wasserstein distance but with reduced computational overhead. For compactly supported measures, Bonnotte [Bonnotte, 2013] showed for instance that the two distances are equivalent. Again, it has been successfully applied in various domains, such as domain adaptation [Lee et al., 2019], texture synthesis and style transfer [Heitz et al., 2021; Elnekave & Weiss, 2022], generative modeling [Deshpande et al., 2018; Wu et al., 2019], regularizing autoencoders [Kolouri et al., 2018], shape matching [Le et al., 2024], and has even been adapted on Riemaniann manifolds [Bonet et al., 2024].

The SW distance between two measures $\mu$ and $\nu$ can be written as the expectation of the one dimensional Wasserstein distance between the projections of $\mu$ and $\nu$ on a line whose direction is drawn uniformly on the hypersphere. It benefits from the simplicity of the Wasserstein distance computation in one dimension. In practice, computing the expectation on the hypersphere is unfeasible, so it is estimated thanks to numerical integration. The most common method for approximating the SW distance is to rely on Monte Carlo approximation, by sampling $M$ random directions uniformly on the hypersphere and approximating the integral by an average on these directions. Since the Wasserstein distance in 1D between two measures of $N$ samples can be computed in $O(N \log N)$, computing this empirical version of Sliced Wasserstein has a runtime complexity of $O(MN \log N)$. This complexity makes it a compelling alternative to the Wasserstein distance, especially when the number $N$ of samples is high.

As a Monte Carlo approximation, the law of large numbers ensures that this empirical Sliced Wasserstein distance converges to the true expectation, with a convergence rate of $O(\frac{1}{\sqrt{M}})$. This convergence speed is slow but independent of the space dimension. However, it is important to keep in mind that to preserve some of the properties of the SW distance, the number $M$ of directions should increase with the dimension. For instance, it has been shown that for the empirical distance to almost surely separate discrete distributions (in the sense that if the distance between two distributions is zero then the two distributions are almost surely equal), the number of directions $M$ must be chosen strictly larger than the space dimension [Tanguy et al., 2023].

Classical Monte Carlo with independent samples is not always optimal, since independent random samples do not cover the space efficiently, creating clusters of points and leaving holes between these clusters. In very low dimension, quadrature rules provide efficient alternative methods to classical Monte Carlo. On the circle for instance, the simplest solution is to replace the $M$ random samples by the roots of unity $\{e^{i\frac{2k\pi}{M}} \mid 0 \leq k \leq M-1\}$: since the function that we wish to integrate is Lipschitz, this ensures that the integral approximation converges at speed $O(\frac{1}{M})$. However, such quadrature rules are unsuitable for high-dimensional problems, as they require an exponential number of samples to achieve a given level of accuracy.

Another alternative sampling strategy is to rely on quasi-Monte Carlo (Q.M.C.) methods, which use deterministic, low-discrepancy sequences instead of independent random samples. Traditional Q.M.C. methods are designed for integration over the unit hypercube $[0,1]^d$. The quality of a Q.M.C. sequence is often measured by its discrepancy, which measures how uniformly the points cover the space. A lower discrepancy correlates with a better approximation, according to the Koksma-Hlawka inequality [Brandolini et al., 2013]. Examples of low-discrepancy sequences for the unit cube include for instance the Halton sequence [Halton, 1964], and the Sobol sequence [Sobol, 1967], and different approaches have been investigated to project such sequences on the hypersphere. While quadrature rules are recommended for very small dimensions ($d = 1$ or 2 for instance), Q.M.C. integration is particularly effective in low to intermediate dimensions. A variant of low-discrepancy sequence is one where randomness is injected in the sequence while preserving its "low discrepancy" property. Such a sequence is called a randomized low-discrepancy sequence, and this is the foundation to randomized quasi-Monte Carlo (R.Q.M.C.) methods [Owen, 2019]. Q.M.C. methods do not only rely on low-discrepancy sequences, but can also use point sets of a given size directly optimized to have low-discrepancy, such as s-Riesz point configurations on the sphere [Götz, 2003]. However Q.M.C. and R.Q.M.C. methods on the sphere have a strong practical downside: they suffer from the curse of dimensionality. Indeed the higher the dimension the harder it is to generate samples with Q.M.C. and R.Q.M.C. approaches. Moreover, the higher the dimension, the slower the convergence rate, and the more regular the integrand needs to be to ensure fast convergence. The recent paper [Nguyen et al., 2024] already proposes an interesting comparison of such Q.M.C. methods to approximate the Sliced-Wasserstein distance in dimension 3, showing that such methods could provide better approximations that conventional M.C. in this specific dimensional setting.

All the sampling strategies mentioned above are designed to provide a good coverage of the space. However, they do not take into account the specific structure of the integrand, which is the Wasserstein distance between the one dimensional projections of the two measures $\mu$ and $\nu$. More involved methods to improve Monte Carlo efficiency include importance sampling, control variates or stratification [Asmussen & Glynn, 2007]. Such variance reduction techniques strategies can also be used in conjunction with quasi-Monte Carlo

integration. Control variates have been explored for Sliced Wasserstein approximation in [Nguyen & Ho, 2023] and [Leluc et al., 2024], showing interesting improvements in intermediate dimensions over classical Monte Carlo.

The goal of this survey is to provide a detailed comparison of these different sampling strategies for the computation of Sliced-Wasserstein in various dimensional settings. It is intended as a user-guide to help practitioners choose the appropriate sampling strategy for their specific problem, depending on the size and dimension of their data, and the type of experiments to be carried out (whether or not they need to compute numerous SW distances for instance). We will also look at the particularities of the different approaches, some being more appropriate than others depending on whether a given level of accuracy is desired (in which case an approach allowing sequential sampling is preferable to one requiring optimization of a point set) or, on the contrary, a given computation time is imposed. We will mainly focus on sampling strategies which are independent of the knowledge of the measures $\mu$ and $\nu$, such as uniform random sampling [Asmussen & Glynn, 2007], orthonormal sampling [Rowland et al., 2019], low-discrepancy sequences mapped on the sphere [Halton, 1964; Sobol, 1967], randomized low-discrepancy sequences mapped on the spheres [Owen, 2019], Fibonacci point sets [Hardin et al., 2016] and Riesz configuration point sets [Götz, 2003]. For the sake of completeness, we also include in our comparison the recent approach [Leluc et al., 2024], which appears to be the most efficient among recent control variates approaches proposed to approximate Sliced Wasserstein.

The paper is organized as follows. Sec. 2 introduces some reminders on the Sliced Wasserstein distance such as its definition and some regularity properties. Sec. 3 explores all the sampling methods considered in this paper, hightlighting their theoretical guarantees, the conditions under which they can be used, and identifying which methods suffer from the curse of dimensionality. Then Sec. 4 provides a comparison of each sampling method's experimental results on different datasets. Finally, in Sec. 5 we offer detailed recommendations for choosing and using these sampling methods effectively in practice.

## 2 Reminders on the Sliced Wasserstein Distance

### 2.1 Definition

In the following, we write $\langle \cdot \mid \cdot \rangle$ the Euclidean inner product in $\mathbb{R}^d$, $\| \cdot \|$ the induced norm, $\mathbb{S}^{d-1} = \{x \in \mathbb{R}^d \mid \|x\| = 1\}$ the unit sphere of $\mathbb{R}^d$. For $\theta \in \mathbb{S}^{d-1}$, we write $\pi_\theta : \mathbb{R}^d \to \mathbb{R}$ the map $x \mapsto \langle \theta | x \rangle$, $s_{d-1}$ the uniform measure over $\mathbb{S}^{d-1}$. We also denote $\#$ the push-forward operation [1].

For two probability measures $\mu$ and $\nu$ supported in $\mathbb{R}^d$ and with finite moments of order 2, the Sliced Wasserstein Distance between $\mu$ and $\nu$ is defined as

$$SW_2^2(\mu, \nu) = \mathbb{E}_{\theta \sim \mathcal{U}(\mathbb{S}^{d-1})}[W_2^2(\pi_\theta \# \mu, \pi_\theta \# \nu)] = \int_{\mathbb{S}^{d-1}} W_2^2(\pi_\theta \# \mu, \pi_\theta \# \nu) ds_{d-1}(\theta). \quad (1)$$

This distance, introduced in [Rabin et al., 2012], has been thoroughly studied and used as a dissimilarity measure between probability distributions in machine learning [Bonneel et al., 2015; Nadjahi, 2021; Kolouri et al., 2018], and more generally as an alternative to the Wasserstein distance. Its simplicity stems from the fact that the Wasserstein distance between two probability measures in one dimension has an explicit formula. Indeed, for two probability measures $\rho_1$ and $\rho_2$ on the line, the Wasserstein distance $W_2(\rho_1, \rho_2)$ can be written

$$W_2^2(\rho_1, \rho_2) = \int_0^1 |F_1^{-1}(t) - F_2^{-1}(t)|^2 dt, \quad (2)$$

where $F_1$ and $F_2$ are the cumulative distribution functions of $\rho_1$ and $\rho_2$, and $F_1^{-1}$ and $F_2^{-1}$ are their respective generalized inverses (see [Santambrogio, 2015] Proposition 2.17). For two one dimensional discrete measures

---

[1] The push-forward of a measure $\mu$ on $\mathbb{R}^d$ by an application $T : \mathbb{R}^d \to \mathbb{R}^k$ is defined as a measure $T\#\mu$ on $\mathbb{R}^k$ such that for all Borel sets $B \in \mathcal{B}(\mathbb{R}^k), T\#\mu(B) = \mu(T^{-1}(B))$.

$\rho_1 = \frac{1}{N} \sum_{k=1}^{N} \delta_{x_k}$ and $\rho_2 = \frac{1}{N} \sum_{k=1}^{N} \delta_{y_k}$, this distance becomes

$$W_2^2(\rho_1, \rho_2) = \frac{1}{N} \sum_{k=1}^{N} |x_{\sigma(k)} - y_{\tau(k)}|^2, \tag{3}$$

where $\sigma$ and $\tau$ are permutations of $[\![1, N]\!]$ which respectively order the sets $\{x_1, \ldots, x_N\}$ and $\{y_1, \ldots, y_N\}$ on the line.

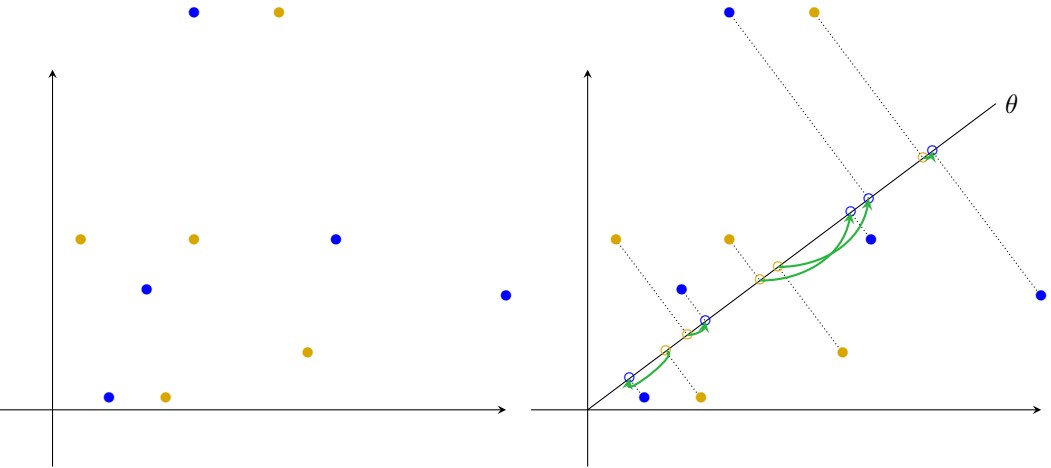

(a) On the left, we can see the two discrete distributions $\mu$ (blue points) and $\nu$ (yellow points). On the right, we have their projections $\pi_\theta \# \mu$ (blue circles) and $\pi_\theta \# \nu$ (yellow circles) along the direction $\theta$. One then takes the increasing ordering of $\pi_\theta \# \mu$ and $\pi_\theta \# \nu$, to obtain the corresponding matchings (green arrows) and computes the cost following Eq. 3.

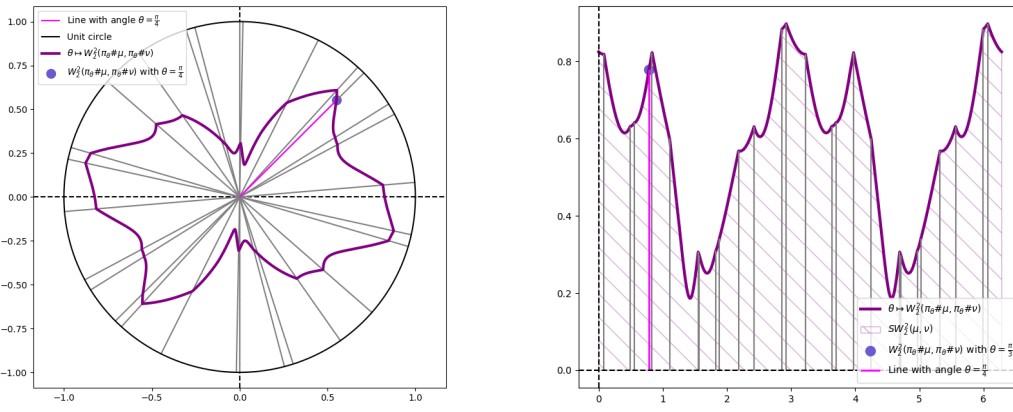

(b) On the left, we have a plot of $\theta \mapsto W_2^2(\pi_\theta \# \mu, \pi_\theta \# \nu)$ in polar coordinates, with the distributions $\mu$ and $\nu$ from Fig. 2.1a (top). The grey lines represent the angles where $\theta \mapsto W_2^2(\pi_\theta \# \mu, \pi_\theta \# \nu)$ is not differentiable, the magenta line is the line of angle $\theta = \frac{\pi}{3}$ and the blue dot is a specific value of $W_2^2(\pi_\theta \# \mu, \pi_\theta \# \nu)$ with the same angle. On the right, we have a $1D$ plot of $\theta \mapsto W_2^2(\pi_\theta \# \mu, \pi_\theta \# \nu)$, here the hashed area represents $SW_2^2(\mu, \nu)$ and again the vertical grey lines represent the values where $\theta \mapsto W_2^2(\pi_\theta \# \mu, \pi_\theta \# \nu)$ is not differentiable.

Figure 2.1: On the first row, Fig. 2.1a illustrates the computation of $W_2^2(\pi_\theta \# \mu, \pi_\theta \# \nu)$ for a fixed $\theta$. On the second row, Fig. 2.1b gives a geometrical illustration of $SW_2^2(\mu, \nu)$ with $\mu, \nu$ taken as in Fig. 2.1a.

As a consequence, the Sliced Wasserstein distance between two discrete probability measures $\mu = \frac{1}{N}\sum\limits_{k=1}^{N}\delta_{x_k}$

and $\nu = \frac{1}{N}\sum\limits_{k=1}^{N}\delta_{y_k}$ on $\mathbb{R}^d$ (i.e. with $(x_k)_{k=1,\dots,N}, (y_k)_{k=1,\dots,N} \in \mathbb{R}^d$) can be rewritten:

$$SW_2^2(\mu,\nu) = \frac{1}{N}\sum_{k=1}^{N}\int_{\mathbb{S}^{d-1}}(\langle x_{\sigma_\theta(k)} - y_{\tau_\theta(k)}, \theta\rangle)^2 ds_{d-1}(\theta) = \frac{1}{N}\sum_{k=1}^{N}\int_{\mathbb{S}^{d-1}}(\langle x_k - y_{\tau_\theta\circ\sigma_\theta^{-1}(k)}, \theta\rangle)^2 ds_{d-1}(\theta), \quad (4)$$

where $\sigma_\theta$ and $\tau_\theta$ denotes respectively permutations which order the one dimensional point sets $(\langle x_k, \theta\rangle)_{k=1,\dots,N}$ and $(\langle y_k, \theta\rangle)_{k=1,\dots,N}$. Fig. 2.1 illustrates the computation of $W_2^2(\pi_\theta\#\mu, \pi_\theta\#\nu)$ for two discrete measures in two dimensions (Fig. 2.1a), and shows how this quantity varies when $\theta$ spans $[0, 2\pi]$ (Fig. 2.1b).

Since the permutations $\sigma_\theta$ and $\tau_\theta$ depends on the direction $\theta$, the integrals in Eq. 1 and Eq. 4 do not have closed forms. For this reason, practitioners rely on Monte Carlo approximations of the form:

$$\frac{1}{NM}\sum_{k=1}^{N}\sum_{j=1}^{M}(\langle x_{\sigma_{\theta_j}(k)} - y_{\tau_{\theta_j}(k)}, \theta_j\rangle)^2, \quad (5)$$

where $\theta_1,\dots,\theta_M$ are i.i.d. and follow a uniform distribution on the sphere. Classically, the convergence rate of such Monte Carlo estimations to SW is $\mathcal{O}(\frac{1}{\sqrt{M}})$ [Hammersley & Handscomb, 1964]. In this context, it is natural to question the optimality of sampling methods to approximate SW efficiently in different scenarios.

## 2.2 Regularity results on $\theta \mapsto W_2^2(\pi_\theta\#\mu, \pi_\theta\#\nu)$

The efficiency of sampling strategies used in numerical integration is highly dependent on the regularity of the functions to be integrated. For this reason, in the following we give some properties of the function (Fig. 2.1b):

$$f : \theta \mapsto W_2^2(\pi_\theta\#\mu, \pi_\theta\#\nu) \quad (6)$$

on the hypersphere $\mathbb{S}^{d-1}$. We first look at classical regularity properties of $f$.

**Proposition 1 :** $f$ is Lipschitz on $\mathbb{S}^{d-1}$.

*Proof.* Let $\mu$ and $\nu$ be two probability measures with finite moments of order 2, and $\theta_1, \theta_2 \in \mathbb{S}^{d-1}$. The triangular inequality on $W_2$ yields

$$|W_2(\pi_{\theta_1}\#\mu, \pi_{\theta_1}\#\nu) - W_2(\pi_{\theta_2}\#\mu, \pi_{\theta_2}\#\nu)| \le W_2(\pi_{\theta_1}\#\mu, \pi_{\theta_2}\#\mu) + W_2(\pi_{\theta_1}\#\nu, \pi_{\theta_2}\#\nu).$$

We also have

$$W_2^2(\pi_{\theta_1}\#\mu, \pi_{\theta_2}\#\mu) = \inf_{X\sim\mu, Y\sim\mu}\mathbb{E}\left[|\langle\theta_1, X\rangle - \langle\theta_2, Y\rangle|^2\right] \le \inf_{X\sim\mu}\mathbb{E}\left[|\langle\theta_1 - \theta_2, X\rangle|^2\right] \le \|\theta_1 - \theta_2\|^2\mathbb{E}_{X\sim\mu}[\|X\|^2].$$

We can show similarly that $W_2^2(\pi_{\theta_1}\#\nu, \pi_{\theta_2}\#\nu) \le \|\theta_1 - \theta_2\|^2\mathbb{E}_{X\sim\nu}[\|X\|^2]$. Thus

$$|W_2(\pi_{\theta_1}\#\mu, \pi_{\theta_1}\#\nu) - W_2(\pi_{\theta_2}\#\mu, \pi_{\theta_2}\#\nu)| \le \|\theta_1 - \theta_2\|\left(\sqrt{\mathbb{E}_{X\sim\mu}[\|X\|^2]} + \sqrt{\mathbb{E}_{X\sim\nu}[\|X\|^2]}\right).$$

$\square$

Since $f$ is Lipschitz continuous, it is differentiable almost everywhere. However the previous result does not give us the set where $f$ is non differentiable. In the following we give a more complete proof when $\mu$ and $\nu$ are discrete following the notations introduced in Sec. 2.1.

**Proposition 2 :** When $\mu$ and $\nu$ are finite discrete measures, $f$ piecewise $\mathcal{C}^\infty$ ($\mathcal{C}_{pw}^\infty$) and Lipschitz on $\mathbb{S}^{d-1}$.

*Proof.* For discrete measures $\mu = \frac{1}{N} \sum_{k=1}^{N} \delta_{x_k}$ and $\nu = \frac{1}{N} \sum_{k=1}^{N} \delta_{y_k}$ on $\mathbb{R}^d$, $f$ can be rewritten as

$$f(\theta) = \min_{\sigma \in \Sigma_N} f_\sigma(\theta), \text{ where } f_\sigma(\theta) = \sum_{k=1}^{N} \langle x_k - y_{\sigma(k)} | \theta \rangle^2, \tag{7}$$

where $\Sigma_N$ is the set of permutations of $[\![1, N]\!]$. We assume that the $\{x_i\}$ (resp. $\{y_j\}$) are all distinct. In the following, we study the regularity of $f$ as a function of $\mathbb{R}^d$ and deduce the regularity properties of its restriction $f_{|\mathbb{S}^{d-1}}$. Observe that each $f_\sigma$ defines a quadratic function on $\mathbb{R}^d$ and $f$, as a minimum of a finite number of such functions, is continuous and also piecewise $\mathcal{C}^\infty$ on $\mathbb{R}^d$. Since $f$ is continuous on $\mathbb{R}^d$, its restriction to $\mathbb{S}^{d-1}$ is also continuous. To show that this restriction to $\mathbb{S}^{d-1}$ is also in $\mathcal{C}_{pw}^\infty$, it is enough to observe that the set of points of $\mathbb{R}^d$ where $f$ is not differentiable is included in the finite union of hyperplanes $\left( \cup_{i,j} \mathrm{Span}(x_i - x_j)^\perp \right) \bigcup \left( \cup_{k,l} \mathrm{Span}(y_k - y_l)^\perp \right)$, since these hyperplanes are the locations where the minimum in Eq. 7 jumps from a permutation $\sigma$ to another one (see Fig. 2.2 as an illustration of those hyperplanes). Each of these hyperplanes intersect $\mathbb{S}^{d-1}$ on a great circle, and we call $\mathcal{U}$ the sphere minus this finite union of great circles. The open set $\mathcal{U}$ (which is dense in $\mathbb{S}^{d-1}$) can be written as the union $\bigcup_{k=1}^{p} V_k$ of a finite number of connected open sets $V_l$, such that on each $V_l$, the permutation $\sigma$ which attains the minimum in Eq. 7 is constant and unambiguous. We write this permutation $\sigma_l$. On each $V_l$, $f_{|\mathbb{S}^{d-1}} = f_{\sigma_l}$, thus is $\mathcal{C}^\infty$ on $V_l$ and its derivative can be obtained as the projection of $\nabla f_{\sigma_l}$ on the hypersphere. For $\theta \in \mathcal{U}$, writing $\sigma_\theta$ the permutation which attains the minimum in Eq. 7 for the direction $\theta$, this derivative can be written

$$\nabla_{(d-1)} f(\theta) = 2 \left( \sum_{k=1}^{N} \left( \langle x_k - y_{\sigma_\theta(k)} | \theta \rangle (x_k - y_{\sigma_\theta(k)}) - \langle x_k - y_{\sigma_\theta(k)} | \theta \rangle^2 \theta \right) \right). \tag{8}$$

Since these derivatives are upper bounded on the compact set $\mathbb{S}^{d-1}$, it follows that $f$ is also Lipschitz on $\mathbb{S}^{d-1}$.

In the case where several $x_i$ (or $y_j$) are equal, several of the functions $f_\sigma$ coincide. For instance, if $x_1 = x_2$, the values of $\sigma(1)$ and $\sigma(2)$ can be exchanged without modifying $f_\sigma$. By eliminating all the redundant functions, we can make the same reasoning as before to show the same regularity results on $f$. In this case, all the pairs $(x_i, x_j)$ with $x_i = x_j$ should be removed when constructing the set of great circles dividing the hypersphere. $\square$

The following proposition will also be useful in the next sections.

**Proposition 3 :** $f \in H^1(\mathbb{S}^{d-1})$, where, for $\alpha \in \mathbb{N}$, the Sobolev space $H^\alpha(\mathbb{S}^{d-1})$ is defined as [Hebey, 1996]

$$H^\alpha(\mathbb{S}^{d-1}) = \{ h \in L^2(\mathbb{S}^{d-1}) \mid \partial^{|j|} h \in L^2(\mathbb{S}^{d-1}), 0 \le |j| \le \alpha \},$$

with $j$ a multi-index and $\partial^{|j|}$ the partial mixed derivative of order $|j|$ on $\mathbb{S}^{d-1}$.

*Proof.* We have seen previously that $f$ is continuous and piecewise $\mathcal{C}^\infty$, piecewise quadratic to be more precise. Thus its weak derivative is piecewise linear with discontinuities on a finite union of hyperplanes, which is $L^2$. $\square$

## 3 Reminders on sampling strategies on the sphere and their theoretical guarantees

In this section, we present the different sampling methods for numerical integration on $\mathbb{S}^{d-1}$ considered in this paper, before comparing them experimentally in Sec. 4. This paper addresses three main types of sampling: random sampling, discrepancy-based sampling, and a control variate approach. The first type includes the classical Monte Carlo (M.C.) method ([Hammersley & Handscomb, 1964], [Lemieux, 2009]) on the sphere and its variant called orthonormal sampling [Rowland et al., 2019]. The second one relies on a concept called the discrepancy ([Lemieux, 2009], [Dick & Pillichshammer, 2010]) of a point set, which represents the number of points in a unit of volume, and can be divided into two categories: low-discrepancy sequences (or digital nets) and point sets (or lattices). Among the former category, we also investigate a

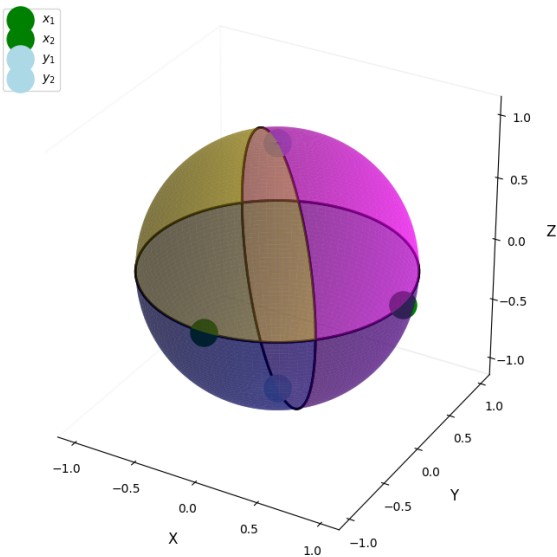

Figure 2.2: Illustration of the open subsets $\bigcup_{k=1}^{p} V_k$ and their intersection with the hyperplanes $\left(\cup_{i,j}\mathrm{Span}(x_i - x_j)^\perp\right) \bigcup \left(\cup_{k,l}\mathrm{Span}(y_k - y_l)^\perp\right)$, in the specific case of two measures made of two diracs $\mu = \frac{1}{2}\sum_{i=1}^{2}\delta_{x_i}$ with $x_1, x_2 = (1,0,0)^T, (0,-1,0)^T$ and $\nu = \frac{1}{2}\sum_{i=1}^{2}\delta_{y_i}$ with $y_1, y_2 = (0,0,1)^T, (0,0,-1)^T$. The hyperplanes divide the sphere into the colored sections where $\sigma_\theta$ and $\tau_\theta$ are constant.

method based on a spherical sliced-Wasserstein type discrepancy [Bonet et al., 2023]. The last type details a control variates method [Lemieux, 2009] using spherical harmonics [Müller, 1998] for this purpose [Leluc et al., 2024]. We will also determine which method, and under which conditions, is theoretically suitable based on the regularity properties established in Sec. 2.2. Tab. 1 presents a taxonomy of all the sampling methods explored in this paper. It details which method's convergence rate result is **independent from the dimension** (i.e. the dimension does not appear in the asymptotic rate), which one can be **computed independently** (i.e. each sample can be generated independently from the others), and which one can be **computed and stored** in advance.

| Sampling types | Dimension independence | Independent computation | Possible pre-computation |
|---|---|---|---|
| **Random Sampling** | | | |
| Uniform Sampling | x | x | x |
| Orthonormal Sampling | x | x | x |
| **Based on discrepancy** | | | |
| Riesz Point Set / Riesz Point Set Randomized | | | x |
| Fibonacci Point Set / Fibonacci Point Set Randomized | | | x |
| Sobol / Sobol Randomized mapped on $\mathbb{S}^{d-1}$ | | x | x |
| Halton / Halton Randomized on $\mathbb{S}^{d-1}$ | | x | x |
| Spherical Sliced Wasserstein Discrepancy | x | | x |
| **Control variates** | | | |
| Spherical Harmonics Control Variates | | | |

Table 1: Taxonomy of the three types of sampling methods investigated in this paper.

Tab. 2 gives a summary of the convergence rate and computational complexity of each sampling method explored in this paper. In this table $n_M = o\left(M^{1/\left(2(d-1)\right)}\right)$.

### 3.1 Random samplings

We first explore classical strategies for randomly generating points on the sphere: uniform sampling [Hammersley & Handscomb, 1964] and orthonormal sampling [Rowland et al., 2019]. These strategies are the

| Sampling types | Theoretical convergence rate | Time complexity | Space complexity |
|---|---|---|---|
| **Random Sampling** | | | |
| Uniform Sampling | $\mathcal{O}(1/\sqrt{M})$ | $\mathcal{O}(M)$ | $\mathcal{O}(M)$ |
| Orthonormal Sampling | None | $\mathcal{O}(M)$ | $\mathcal{O}(M)$ |
| **Based on discrepancy** | | | |
| Riesz Point Set / Riesz Point Set Randomized | $1/M$ on $\mathbb{S}^1$, Not applicable otherwise | $\mathcal{O}(M^2)$ | $\mathcal{O}(M)$ |
| Fibonacci Point Set / Fibonacci Point Set Randomized | Not applicable | $\mathcal{O}(M)$ | $\mathcal{O}(M)$ |
| Sobol / Sobol Randomized mapped on $\mathbb{S}^{d-1}$ | None | $\mathcal{O}(M\log_b^2(M))$ | $\mathcal{O}(M)$ |
| Halton / Halton Randomized on $\mathbb{S}^{d-1}$ | None | $\mathcal{O}(M\log_b^2(M))$ | $\mathcal{O}(M)$ |
| Spherical Sliced Wasserstein Discrepancy | None | $\mathcal{O}(M\log(M))$ | $\mathcal{O}(M)$ |
| **Control variates** | | | |
| Spherical Harmonics Control Variates | $\mathcal{O}\big(1/(n_M\sqrt{M})\big)$ | $\mathcal{O}(M)$ | $\mathcal{O}(M)$ |

Table 2: Convergence rate, time complexity and spacial complexity (w.r.t the sampling number) summary of the sampling methods studied in this paper.

most commonly used for estimating $SW_2^2$, and their convergence rates do not depend on the dimension of the input measures.

### 3.1.1 Classical Monte Carlo

The classical Monte Carlo method uses uniform random sampling to generate the projection angles. For $(\theta_M)_{M\in\mathbb{N}^*}$ i.i.d. samples of $s_{d-1}$ [2], we write the Monte Carlo Estimator

$$X_M := \frac{1}{M}\sum_{i=1}^{M} f(\theta_i) \text{ with } M\in\mathbb{N}^*. \tag{9}$$

The law of large numbers ensures that $X_M$ converges a.s. to $SW_2^2(\mu,\nu) = \mathbb{E}_{\theta\sim s_{d-1}}[f(\theta)]$ as $M$ goes to infinity. Moreover, the rate of convergence for this unbiased estimator is given by

$$\sqrt{\mathbb{V}[X_M]} = \sqrt{\frac{\mathbb{V}[X_1]}{M}} = \frac{\sigma}{\sqrt{M}}, \tag{10}$$

where $\sigma^2 = \mathbb{V}[f(\theta)] = \int_{\mathbb{S}^{d-1}} f^2(\theta)ds_{d-1}(\theta) - SW_2^4(\mu,\nu) < +\infty$. This convergence rate in Eq. 10 does not depend on the dimension of the input measures. In order to derive confidence intervals for $SW_2^2(\mu,\nu)$, we can rely on the Central Limit Theorem [Fischer, 2010] , which states that

$$\sqrt{M}\frac{[X_M - SW_2^2(\mu,\nu)]}{\sigma} \xrightarrow[M\to+\infty]{\mathcal{L}} \mathcal{N}(0,1),$$

This allows us to compute confidence intervals for $SW_2^2(\mu,\nu)$ by using the quantiles of the standard normal distribution. This means that for $M$ large enough, $\mathbb{P}\left(X_M - SW_2^2(\mu,\nu) \in \left[-\frac{\sigma q_{1-\alpha/2}}{\sqrt{M}}, \frac{\sigma q_{1-\alpha/2}}{\sqrt{M}}\right]\right) \xrightarrow[M\to+\infty]{} 1 - \alpha$, with $\alpha$ in $[0,1]$ and $q_{1-\alpha/2}$ the quantile of level $1-\alpha/2$ of $\mathcal{N}(0,1)$. One strategy for choosing $M$ is taking $M$ such that $\frac{\sigma q_{1-\alpha/2}}{\sqrt{M}} \leq \varepsilon$ with $\varepsilon \geq 0$ a chosen precision. The value of $\sigma$ being unknown, a possibility is to plug a consistent estimator of $\sigma^2$, such as

$$\hat{\sigma}_M^2 = \frac{1}{M}\left[\sum_{i=1}^{M} f(\theta_i)^2 - X_M^2\right].$$

Xianliang & Zhongyi [2022] provide an alternative criteria for choosing $M$, however it is quite impractical as it requires to compute the Wasserstein distance between $\mu$ and $\nu$.

---

[2]In practice, to simulate a random variable $\theta \sim s_{d-1}$, one takes a normal random variable $Z \sim \mathcal{N}(0, I_d) \neq 0$ and chooses $\theta = \frac{Z}{\|Z\|} \sim s_{d-1}$ [Asmussen & Glynn, 2007].

### 3.1.2 Orthonormal sampling

A variant of the uniform sampling covered in Sec. 3.1.1 was introduced by [Rowland et al., 2019], which presents a simple variant for the previous Monte Carlo estimator $X_M$ by sampling random orthonormal bases. This method is inspired by variance reduction techniques known as stratification [Lemieux, 2009]. Let $O(d)$ be the orthogonal group in $\mathbb{R}^d$. For $(\Theta_P)_{P \in \mathbb{N}^*} \sim \mathcal{U}\big(O(d)\big)$, denoting $\theta_1, \ldots, \theta_M$ all the columns of the matrices $\Theta_1, \ldots, \Theta_K$, we define $Y_M = \dfrac{1}{M} \sum_{i=1}^{M} f(\theta_i)$. It is easy to show that each $\theta_i$ follows the uniform distribution on $\mathbb{S}^{d-1}$ [Rowland et al., 2019]. As a consequence, the estimator $Y_M$ is still unbiased. Although it is not possible to show that $Y_M$ has a smaller variance than $X_M$ in general, this estimator is most of the time more efficient than $X_M$ in our experiments and show an equivalent or better rate of convergence in practice. This might be due to the fact that the diversity of the samples is increased by the orthonormality constraint.

**Remark 1 :** Other fully random point processes on $[0,1]^2$ or $\mathbb{S}^2$ suitable for Monte Carlo integration are studied in the literature. Among them, we can mention Determinantal Point Process (DPP). Recent works, such as [Feng et al., 2023], have proposed DPP methods directly on the sphere $\mathbb{S}^2$. Unfortunately, due to the lack of publicly available implementations, we could not experiment efficiently with these methods.

## 3.2 Sampling strategies based on discrepancy

We examine in this section two different types of deterministic sampling based on discrepancy: low-discrepancy sequences (digital nets) and low-discrepancy point sets (lattices). They were developed to replace random sampling, expecting to have a better convergence rate than the classical Monte Carlo method.

### 3.2.1 Low-discrepancy sequences

Quasi-random sequences, better known as low-discrepancy sequences (L.D.S.), are sequences mimicking the behavior of random sequences while being entirely deterministic. To date, these sequences are only defined on the unit hypercube $[0,1]^d$. We introduce below a first definition of discrepancy ([Lemieux, 2009],[Dick & Pillichshammer, 2010]).

**Definition 1 :** The discrepancy of a set of points $P = \{u_1, \ldots, u_M\}$ in $[0,1]^d$ is defined as

$$D_M(P) = \sup_{I \in \mathcal{I}} \left| \frac{|P \cap I|}{M} - \lambda^{\otimes d}(I) \right|,$$

where $|A|$ denotes the cardinal of a set $A$, $\lambda^{\otimes d}$ is the $d$-dimensional Lebesgue measure and $\mathcal{I} = \big\{ \prod_{i=1}^{d} [a_i, b_i[ \mid 0 \le a_i < b_i \le 1\big\}$. The star-discrepancy $D_M^*(P)$ is defined the same way with $\mathcal{I}^* = \big\{ \prod_{i=1}^{d} [0, b_i[ \mid 0 \le b_i \le 1\big\}$.

We can now provide a definition of a Low discrepancy sequence (L.D.S.).

**Definition 2 :** Let $(u_m)_{m \in \mathbb{N}^*}$ be a sequence in $[0,1]^d$. Denoting $P_M = \{u_1, \ldots, u_M\}$ for any $M \in \mathbb{N}^*$, $u$ is a L.D.S. if

$$D_M^*(P_M) \xrightarrow[M \to +\infty]{} 0.$$

The notion of discrepancy is important because it is related to the error made when approximating an integral on the hypercube by its Monte Carlo approximation. This relation is made explicit by the Koksma-Hlawka inequality ([Lemieux, 2009]; [Dick & Pillichshammer, 2010]; [Brandolini et al., 2013]). This inequality requires to introduce the notion of Hardy-Krause variation $V_h$ of a function $h$ on $[0,1]^d$ [Aistleitner et al., 2016], which is out of the scope of this paper, but can be broadly understood as a measure of the oscillation of $h$ on the unit cube $[0,1]^d$.

**Proposition 4** (Koksma-Hlawka inequality) **:** Let $h : [0,1]^d \to \mathbb{R}$ have bounded variation $V_h$ on $[0,1]^d$ in the sense of Hardy-Krause [Aistleitner et al., 2016]. Then for $\{u_1, \ldots, u_M\}$ a point set in $[0,1]^d$, we have

$$\left| \frac{1}{M} \sum_{k=1}^{M} h(u_k) - \int_S h(x) d\lambda^{\otimes d}(x) \right| \leq V_h D_M^*(u_1, \ldots, u_M). \tag{11}$$

The proof of this inequality and basic results on discrepancy theory can be found in[Kuipers & Niederreiter, 2012] and [Dick & Pillichshammer, 2010]. Eq. 11 shows that the absolute error made by the Monte Carlo approximation is upper bounded by a term depending only on $h$ and the star discrepancy. Compared to the Central Limit Theoreom, this inequality is not probabilistic and not asymptotic, the bound being valid for every $M \in \mathbb{N}^*$. An important limitation is the term $V_h$, which is impractical to compute directly. When $d = 1$, this term is exactly the total variation of $h$, but in general, it is only upper bounded by the total variation. In the case of our function $f$ involved in the estimation of $SW$, $V_f < +\infty$ holds since $f$ is Lipschitz continuous. Another limitation of the previous bound is that the rate of convergence of the star discrepancy $D_M^*$ of a sequence is most of the time not explicit and difficult to compute [Owen, 2005].

Nevertheless, this proposition ensures that if the rate of convergence of the star discrepancy of a sequence is better than $O(\frac{1}{\sqrt{M}})$, for $M$ large enough the approximation of the quasi Monte Carlo approximation using this sequence will outperform the one of classical Monte Carlo.

In the following, we present two L.D.S. defined on the unit square $[0,1]^d$, and see how their star discrepancy decreases with $M$. We then focus on practical methods to map these sequences from the hypercube to the hypersphere $\mathbb{S}^{d-1}$.

### 3.2.1.1 Halton sequence

The Halton sequence $(u_i)_{i \in \mathbb{N}} \in (\mathbb{R}^d)^{\mathbb{N}}$ [Halton, 1964] is a generalization of the von der Corput sequence [van der Corput, 1935]. In the following, we write, for any integer $i$, $c_l(i)$ the coefficients from the expansion of $i$ in base $b$, and we define the radical-inverse function in base $b$ as

$$\phi_b(i) = \sum_{l=0}^{+\infty} c_l(i) b^{-l-1}, \forall i \in \mathbb{N}.$$

The Halton sequence in dimension $d$ is then defined as

$$u_i = (\phi_{b_1}(i), \ldots, \phi_{b_d}(i))^T,$$

where $b_i$ is chosen as the i-th prime number.

### 3.2.1.2 Sobol sequence

This sequence uses the base $b = 2$. To generate the $j$-th coordinate of the $i$-th point $u_i$ in a Sobol sequence [Sobol, 1967], one needs a primitive polynomial of degree $n_j$ in $\mathbb{Z}/2\mathbb{Z}[X]$,

$$X^{n_j} + a_{1,j} X^{n_j-1} + a_{2,j} X^{n_j-2} + \ldots + a_{n_j-1,j} X + 1.$$

This polynomial is used to define a sequence of positive integers $(m_{k,j})$ by recurrence, with $+_{\mathbb{Z}/2\mathbb{Z}}$ the inner law of $\mathbb{Z}/2\mathbb{Z}$:

$$m_{k,j} = 2a_{1,j} m_{k-1,j} +_{\mathbb{Z}/2\mathbb{Z}} 2^2 a_{2,j} m_{k-2,j} +_{\mathbb{Z}/2\mathbb{Z}} \ldots +_{\mathbb{Z}/2\mathbb{Z}} 2^{n_j} m_{k-n_j,j} +_{\mathbb{Z}/2\mathbb{Z}} m_{k-n_j,j}.$$

The values $m_{k,j}$, for $1 \leq k \leq n_j$, can be chosen chosen arbitrarily provided that each is odd and less than $2^k$. Then one generates what is called direction numbers:

$$v_{k,j} = \frac{m_{k,j}}{2^k}.$$

The $j$-th coordinate of $u_i$ is then obtained as

$$u_{i,j} = \sum_{k=1}^{+\infty} c_k(i) v_{k,j}.$$

**3.2.1.3   Convergence rate of Halton and Sobol sequences**   Both sequences (Halton and Sobol) have a star discrepancy which converges to 0 (which means that they are indeed L.D.S.). The convergence rate is given by the following property [Niederreiter, 1988] [Owen, 2019].

**Proposition 5 :** Let $(u_m)_{m\in\mathbb{N}^*}$ be either the Halton sequence or Sobol sequence in $[0,1]^d$. Then for $M \geq 1$, we have

$$D_M^*(u_1, \ldots, u_M) \leq c_d \frac{log(M)^d}{M}$$

where $c_d$ is a constant that depends only on the dimension.

Thanks to Eq. 11, for any function $h$ such that $V_h < +\infty$ (which is the case for our function $f$), this implies a convergence rate of the Monte Carlo estimator using these sequences in $\mathcal{O}(\frac{log(M)^d}{M})$, which means $\mathcal{O}(M^{-1+\epsilon})$ for every $\epsilon > 0$. This convergence rate is better than the one of classical Monte Carlo with i.i.d. sequences, even if the rate of convergence slows down when the dimension increases, because of the term $log(M)^d$.

**Remark 2 :** Note that L.D.S. are designed to mimic the behavior of a random uniform sampling in $[0,1]^d$ while being completely deterministic. This deterministic behavior leads to patterns in the sampling; because of those patterns, the higher the dimension, the harder it is for those to fill the "gaps" in $[0,1]^d$. Moreover, the term $log(M)^d$ implies that one needs $M$ to be very large (exponential) to get the same level of space coverage in high dimension than in low dimension.

**Remark 3 :** Observe that both for Sobol and Halton sequences, generating $M$ values has a complexity in $\mathcal{O}(Mlog_b^2(M))$, where $b$ is the base (or smallest basis for Halton) chosen.

**3.2.1.4   L.D.S. on the sphere**
To our knowledge, there is no true L.D.S. on the unit sphere $\mathbb{S}^{d-1}$ for $d \geq 3$, this question remaining an active research area. Practitioners typically map L.D.S. from the hypercube to the hypersphere, using one of the methods described below:

- **Equal area mapping** [Aistleitner et al., 2012]: this method is only defined for mapping points in the unit square to $\mathbb{S}^2$. Denoting $(z_1, z_2) \in [0,1[^2$, one gets a point $u = \Phi(2\pi z_1, 1 - 2z_2)$ on $\mathbb{S}^2$ with:

$$\Phi(\eta, \beta) = \left( \sqrt{1-\beta^2}\cos(\eta), \sqrt{1-\beta^2}\sin(\eta), \beta \right), \ \eta, \beta \in [0,1[. \tag{12}$$

- **Spherical coordinates** [Arfken et al., 2011]: This method maps the points from an L.D.S. in $[0,1]^{d-1}$ to $\mathbb{S}^{d-1}$ by using the spherical coordinates. Unfortunately, we found that the resulting sampling is usually not competitive compared to other sampling methods.

- **Normalization onto the sphere** [Basu, 2016]: An L.D.S. is generated in the $d$-hypercube $[0,1]^d$ and mapped to $\mathbb{R}^d$ using the inverse cumulative distribution function of the standard normal distribution (separately on each dimension). Then each point in the resulting sequence is normalized by its norm to map it onto $\mathbb{S}^{d-1}$.

**Specific case of $\mathbb{S}^2$.**
In the specific case of $\mathbb{S}^2$, it has been shown by Aistleitner et al. [2012] that if $u$ is an L.D.S in $[0,1]^2$ and $\Phi$ the equal area mapping defined in Eq. 12, the spherical cap discrepancy $D_{\mathbb{L}_2,M}(\Phi(P))$ (see definition 3 in the next section) of the mapped sequence is in $\mathcal{O}(\frac{1}{M^{1/2}})$. However, their experiments showed that the correct order seems rather to be $\mathcal{O}\left(\frac{log^c(M)}{M^{3/4}}\right)$ for $1/2 \leq c \leq 1$.

### 3.2.2 Deterministic point sets on $\mathbb{S}^{d-1}$

This section details different methods to design well distributed point sets on $\mathbb{S}^{d-1}$. Contrary to the L.D.S. defined above, these point sets are defined directly on the sphere, in order to be approximately uniformly distributed on $\mathbb{S}^{d-1}$. To measure this uniformity, we can rely on the notion of spherical cap on the sphere: a spherical cap of center $c \in \mathbb{S}^{d-1}$ and $t \in [-1,1]$ is defined as

$$C(c,t) = \{x \in \mathbb{S}^{d-1} \mid \langle x,c \rangle > t\}. \tag{13}$$

In other words, a spherical cap is the intersection of a portion of the sphere and a half-space (see Fig. 3.1 for an illustration).

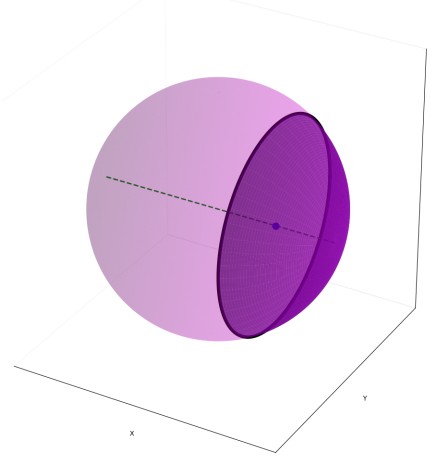

Figure 3.1: Illustration of a spherical cap on $\mathbb{S}^2$. The circle represents the intersection of the plane $\langle x,c \rangle = t$ with the sphere, and the purple colored area is the cap $C(c,t)$ as noted in Eq. 3.2.2.

To the best of our knowledge, there is no equivalent to the Koksma-Hlawka inequality for the sphere in full generality [Brauchart, 2011]. A sequence of points $\{u_n\}$ on $\mathbb{S}^{d-1}$ is said asymptotically uniformly distributed on $\mathbb{S}^{d-1}$ if for every spherical cap $C$, the proportion of points inside the cap, converges to the measure of the cap $s_{d-1}(C)$. It can be shown that this assumption is equivalent to assume that for every continuous function $h$, the Monte Carlo approximation $\frac{1}{M}\sum_{k=1}^{M} h(u_k)$ converges to $\mathbb{E}_{\theta \sim s_{d-1}}[h(\theta)]$.

In order to get a non asymptotic notion of the uniformity of a point set on $\mathbb{S}^{d-1}$, we can rely on different notions of spherical cap discrepancy on the sphere, defined as follows.

**Definition 3 :** The spherical cap max-discrepancy of a point set $P_M$ of size $M$ is defined as [Marzo & Mas, 2021]:

$$D_{max}(P_M) = \sup_{c \in \mathbb{S}^{d-1}, t \in [-1,1]} \left\{ \left| \frac{|P_M \cap C(c,t)|}{M} - s_{d-1}\big(C(c,t)\big) \right| \right\}.$$

The spherical cap $\mathbb{L}_2$-discrepancy of a point set $P_M$ of size $M$ is defined as [Brauchart, 2011]:

$$D_{\mathbb{L}_2}^2(P_M) = \left\{ \int_{-1}^{1} \int_{\mathbb{S}^{d-1}} \left| \frac{|P_M \cap C(c,t)|}{M} - s_{d-1}\big(C(c,t)\big) \right|^2 ds_{d-1}(c)dt \right\},$$

where $C(c,t)$ is a spherical cap of center $c$ and height $t$.

Again, the idea is to compare the proportion of points in $P_M$ that fall inside a spherical cap with the measure of the cap. This comparison is done for all possible caps on the sphere, and $D_{max}$ represents the worst error over all possible caps, while $D_{\mathbb{L}_2}^2$ represents the average squared error over all possible caps.

When using Q.M.C. on the hypersphere to approximate the integral of functions $h$, another notion often used in the literature is the worst-case (integration) error (W.C.E.) on a Banach space of functions, which is the largest possible error made by the method on the space. For instance, on $H^\alpha(\mathbb{S}^{d-1})$,

**Definition 4 :** For $P_M = \{u_1, \ldots, u_M\}$, for $\alpha \in \mathbb{N}$

$$WCE\left(P_M, H^\alpha(\mathbb{S}^{d-1})\right) = \sup_{h \in H^\alpha(\mathbb{S}^{d-1})} \left| \frac{1}{M} \sum_{m=1}^M h(u_m) - \frac{1}{s_{d-1}(\mathbb{S}^{d-1})} \int_{\mathbb{S}^{d-1}} h(w)ds_{d-1}(w) \right|.$$

Under some regularity condition, a sufficient and necessary one being $\alpha \geq \frac{1}{2} + \frac{d-1}{2}$ for $H^\alpha(\mathbb{S}^{d-1})$, Brauchart & Dick [2013] show that optimizing the spherical cap $\mathbb{L}_2$-discrepancy is equivalent to optimizing the W.C.E. thanks to the Stolarsky's invariant principle [Stolarsky, 1973]. In the case of our function $f$, we have seen that $f$ is regular enough in the specific case of $\mathbb{S}^1$, since $f \in H^\alpha(\mathbb{S}^1)$ with $\alpha = 1 = \frac{1}{2} + \frac{1}{2}$. However in dimension larger than 3, this result does not hold anymore since $f$ does not belong to any Sobolev space $H^\alpha(\mathbb{S}^d)$ with $\alpha > 1$.

### 3.2.2.1 Fibonacci point set on $\mathbb{S}^2$

Denoting $\varphi$ the polar angle and $\chi$ the azimuthal angle forming the geographical coordinates $(\varphi, \chi)$, we retrieve the Cartesian coordinates $(x, y, z)$ using the spherical coordinates (see Fig. 3.2 for an illustration). Noting $\phi = \frac{1+\sqrt{5}}{2}$ the golden ratio, the $m$-th point $u_m = (\varphi_m, \chi_m)$ of the Fibonacci point set is given by

$$\varphi_m = \arccos\left(\frac{2m}{2M+1}\right),$$

$$\chi_m = 2m\pi\phi^{-2}.$$

It is a simple and efficient way, convergence rate wise, to generate points on $\mathbb{S}^2$ for the quasi-Monte Carlo method but it is only defined on $\mathbb{S}^2$. The complexity of the generation is linear in $M$, and according to [Marques, 2013], the corresponding convergence rate for the W.C.E. and the $\mathbb{L}_2$-spherical cap discrepancy is in $\mathcal{O}(\frac{1}{M^{3/4}})$. For an extensive list of other popular point configurations on $\mathbb{S}^2$, see [Hardin et al., 2016].

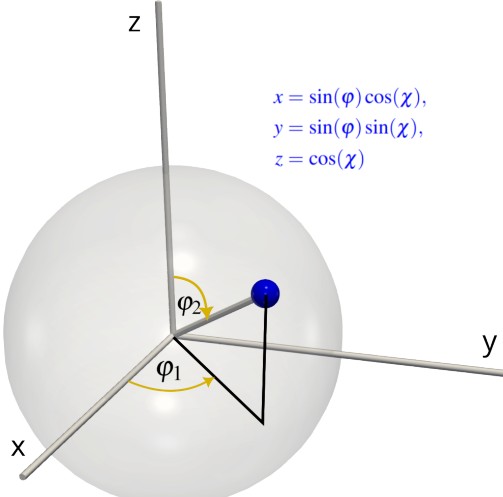

Figure 3.2: Illustration of the spherical coordinates in $\mathbb{R}^3$ for points on the sphere $\mathbb{S}^2$.

### 3.2.2.2 Equi-distributed points generated by the discrete s-Riesz energy

Another classical way to define equi-distributed point sets on the hypersphere is to rely on optimization. In such methods, the point set $P_M$ is defined as the minimizer of a certain energy functional $E_s$,

$$P_M^* := \arg\min_{u_1, \ldots, u_M \in \mathbb{S}^{d-1}} E_s(u_1, \ldots, u_M).$$

The most common energy functional is the s-Riesz energy, which is defined as follows.

**Definition 5 :** For $s \geq 0$ and $P_M = \{u_1, \ldots, u_M\}$ a set of points on $\mathbb{S}^{d-1}$, the s-Riesz energy of $P$ is defined as

$$
E_s(P_M) = \begin{cases} \displaystyle\sum_{i \neq j} \frac{1}{\|u_i - u_j\|^s} & \text{if } s > 0, \\ \displaystyle\sum_{i \neq j} \log \frac{1}{\|u_i - u_j\|} & \text{if } s = 0. \end{cases}
$$

The resulting point set is called a minimal $s$-energy configuration. The s-Riesz energy can also be defined for $s < 0$, in this case the point set $P_M$ is obtained as the maximizer of $E_s = \displaystyle\sum_{i \neq j} \|u_i - u_j\|^s$ [Brauchart, 2011]. Minimising $E_s$ is non trivial, the functional being not convex, and the problem becomes more complex when the dimension increases. Minimal energy configuration points for $E_s$ are called Fekete points and it is known that for $0 \leq s < d$, these sets are asymptotically uniformly distributed with respect to the normalized surface measure $s_{d-1}$, which means that Monte Carlo estimates using the Fekete points converge to the integral against $s_{d-1}$ [Marzo & Mas, 2021].

The spherical cap $\mathbb{L}_2$-discrepancy of a point configuration is minimal if and only if the sum of distances in the configuration is maximal. This would correspond to maximizing a s-Riesz energy for $s = -1$ [Brauchart, 2011]. However, the link between the configurations of minimal $s$-Riesz energy and the max or $\mathbb{L}_2$ discrepancies of these configurations is in general not straightforward, see [Brauchart, 2011], [Marzo & Mas, 2021], [Götz, 2003]. For $0 \leq s < d$, and $P_M$ a minimizer of size $M$ of the Riesz s-energy on $\mathbb{S}^{d-1}$, the authors of [Marzo & Mas, 2021] show that

$$
D_{max}(P_M) \lesssim O\left(\max\left(M^{-\frac{2}{d(d-s+1)}}, M^{-\frac{2(d-s)}{d(d-s+4)}}\right)\right).
$$

This implies that $D_{max}(P_M) \xrightarrow[M \to +\infty]{} 0$, but the speed of convergence degrades with the dimension $d$, which means that the uniformity of these configurations is likely to suffer from the curse of dimensionality. Fig. 3.4 shows an example of s-Riesz points and Fibonacci points on $\mathbb{S}^2$ with 500 points.

**Remark 4 :** Since computing Riesz point configurations involves optimization (with a non linear complexity), the time needed to generate those points can be impractical. Note that generally the generation of the $s$-Riesz configuration points has a runtime complexity of $\mathcal{O}(TM^2)$, where $T$ is the number of iterations of the optimization loop.

**In the specific case of** $S^1$, the Fekete points are unique up to a rotation, and are the $M$-th unit roots (see [Götz, 2003] and see Fig. 3.3 for an illustration):

$$
\left\{ e^{\frac{2ik\pi}{M}} \mid k = 0, \ldots, M - 1 \right\}.
$$

This explains why for 2D discrete measures, a uniform grid on $\mathbb{S}^1$ gives better results than any other sampling method for computing $SW_2^2$, as we will see in Sec. 4.

### 3.2.3 Random Quasi Monte-Carlo

The principle of Randomized Quasi-Monte Carlo (R.Q.M.C.) methods is to reintroduce stochasticty in Q.M.C. sequences. Indeed, Q.M.C. methods such as the ones described in Sections Sec. 3.2.1 and Sec. 3.2.2 are deterministic. For a given $M$, the estimator given by one of these methods is always the same. As such, we cannot easily estimate the error or the variance of the Monte Carlo approximation. Besides, while results such as the Koksma-Hlawka inequality ensures that they converge at a certain rate, the different quantities involved in the inequality are much more complex to estimate than the one involved in the Central Limit theorem. Random Quasi-Monte Carlo methods were especially designed to recover this ability to estimate the error easily. These sequences are usually defined on $[0, 1]^d$.

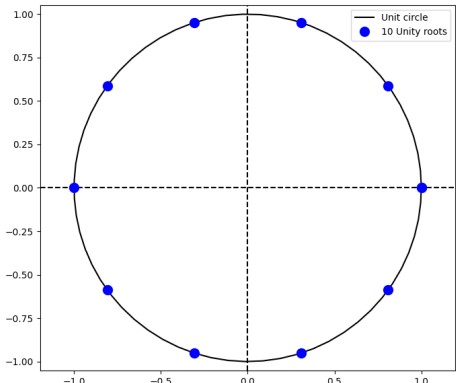

Figure 3.3: Plot of the 10-th unity roots, i.e solutions to the equation $z^{10} = 1$.

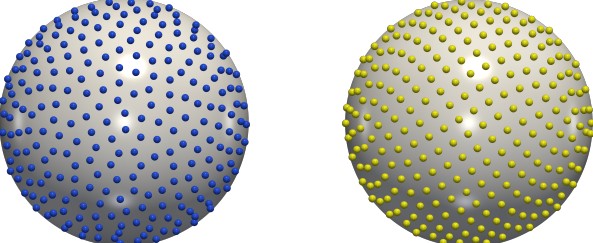

Figure 3.4: Illustration of s-Riesz points (on the left) and Fibonacci points (on the right) on $S^2$, with 500 points for both configurations.

**Definition 6** ([Owen, 2019]) : Let $\{\hat{u}_i\}_{i \geq 1}$ be a sequence of points in $[0,1]^d$. It is said to be suitable for R.Q.M.C. if $\forall i$, $\hat{u}_i \sim \mathcal{U}([0,1]^d)$ and if there exist a finite $c > 0$ and $K > 0$ such that for all $M \geq K$,

$$\mathbb{P}\left[D_M^*(\hat{P}_M) < c\frac{log^d(M)}{M}\right] = 1, \text{ where } \hat{P}_M = \{\hat{u}_1, \ldots, \hat{u}_M\}.$$

Denoting $X_M = \dfrac{1}{M}\sum_{i=1}^{M} h(\hat{u}_i)$ the empirical estimator of $\mathbb{E}_{\theta \sim s_{d-1}}[h(\theta)]$, the assumption $\hat{u}_i \sim \mathcal{U}([0,1]^d)$ implies that $X_M$ is unbiased. Besides, the previous inequality implies that if $\{\hat{u}_i\}_{i \geq 1}$ is suitable for R.Q.M.C., then the variance of $X_M$ is bounded by $c^2 V_h^2 \frac{log^{2d}(M)}{M^2}$. For functions $h$ such that $V_h < \infty$, this yields a convergence rate in $\mathcal{O}\left(log^d(M)/M\right)$, similar to the one of low discrepancy sequences.

Once a randomization method is chosen (such that it provides suitable R.Q.M.C. sequences), the process can be repeated several times to obtain $K$ independent random estimators $X_M^1, \ldots X_M^K$ of $\mathbb{E}_{\theta \sim s_{d-1}}[h(\theta)]$. The agregated estimate $X_{M,K} = \dfrac{1}{K}\sum_{k=1}^{K} X_M^K$ has a variance decreasing in $O\left(log^d(M)/(MK^{-1/2})\right)$. One of the key advantages of this approach is that this variance (or confidence intervals) can be estimated by the empirical variance of the $K$ independent estimators.

There are several ways to generate sequences from low discrepancy sequences on $[0,1]^d$ in order to make them suitable for R.Q.M.C.. One of the most simple methods consists in applying the same random shift $U$ to all points in the sequence, and taking the result modulo 1 componentwise [Lemieux, 2009]. More involved methods, such as Digital shift or Scrambling, are described in [Lemieux, 2009] and [Owen, 2019].

However, to the best of our knowledge, there is no proper R.L.D.S. on the sphere, as stated by Nguyen et al. [2024]. In practice, R.L.D.S. on the unit cube are mapped onto the sphere by the methods described in paragraph 3.2.1.4. Another possibility, as done in Nguyen et al. [2024], is to generate a random rotation matrix and apply it directly on point configurations on $\mathbb{S}^{d-1}$, such as the ones described in Sec. 3.2.2.

### 3.3 Spherical Sliced Wasserstein

A sampling method based on a Sliced-Wasserstein type discrepancy on the sphere $\mathbb{S}^{d-1}$ was developed by Bonet et al. [2023] for $d \geq 3$. We denote $\mathbb{C}_{d,2}$ the set of great circles of $\mathbb{S}^{d-1}$, a great circle being the intersection between a plane of dimension 2 and $\mathbb{S}^{d-1}$ [Jung et al., 2012]. The authors of Bonet et al. [2023] define a pseudo distance, called Spherical Sliced Wasserstein distance, between two probability measures $\Theta, \Xi$ defined on $\mathbb{S}^{d-1}$:

$$SSW_2^2(\Theta, \Xi) = \int_{\mathbb{C}_{d,2}} W_2^2(\pi_C \# \Theta, \pi_C \# \Xi) d\zeta(C), \tag{14}$$

where for all $x \in \mathbb{S}^{d-1}, \pi_C(x) = \arg\min_{y \in C} d_{\mathbb{S}^{d-1}}(x, y)$ with $d_{\mathbb{S}^{d-1}}(x, y) = \arccos(\langle x, y \rangle)$ [Fletcher et al., 2004] and $\zeta$ is the uniform distribution over $\mathbb{C}_{d,2}$.

As shown in Bonet et al. [2023], this distance can be used to sample points on $\mathbb{S}^{d-1}$ by minimizing $SSW_2$ between a discrete measure $\Theta = \frac{1}{M} \sum_{i=1}^{M} \delta_{\theta_i}$ and the uniform measure $\Xi = s_{d-1}$ on $\mathbb{S}^{d-1}$. To this aim, for $C_1, \ldots, C_L$ L independent great circles, they approximate $SSW_2^2(\Theta, \Xi)$ by its Monte Carlo approximation $Z_L(\Theta, \Xi) = \frac{1}{L} \sum_{l=1}^{L} W_2^2(\pi_{C_l} \# \Theta, \pi_{C_l} \# \Xi)$. Then, they note that $\pi_{C_l} \# s_{d-1} = s_1$ [Jung, 2021] for each $l$, and derive a closed form for $W_2^2(\pi_{C_l} \# \Theta, s_1)$ based on Delon et al. [2010]. The final distance $SSW_2^2(\Theta, \Xi)$ can then be optimized with respect to the point positions $\theta_i$ with a projected gradient descent.

**Remark 5 :** There are cases in which $SSW$ is a metric:

- Based on Quellmalz et al. [2023], $SSW$ is a metric between any two probability measures on $\mathbb{S}^2$.
- A result from Liu et al. [2024] also shows that $SSW$ is a metric between any two absolutely continuous probability measures with continuous density functions on $\mathbb{S}^{d-1}$ for $d \geq 3$.

**Remark 6 :** Noting $T$ the number of iterations for the gradient descent algorithm, and $L$ as above, then the time complexity of this method is in $\mathcal{O}(TLM\log(M))$.

**Remark 7 :** Notice that $SSW$'s form is similar to the $\mathbb{L}_2$-spherical cap discrepancy, where instead of averaging the "error" made by the sampling on a spherical cap, it averages the "error" made by the sampling on a great circle.

### 3.4 Variance reduction

All methods described so far are based on the idea of generating points on the sphere in such a way that these points are sufficiently well distributed to be used for Monte Carlo integration, and ideally yield faster convergence than M.C. with i.i.d. sequences. These point sequences or point sets are defined independently of the function to be integrated.

More involved approaches, such as importance sampling or control variates, use the knowledge of the function to be integrated to improve Monte Carlo estimators by decreasing their variance. Recently, two control variates based methods have been developed to estimate the Sliced Wasserstein distance. A control variate is a centered random vector $Y \in \mathbb{R}^p$, easy to sample, with finite second moments. Assume we want to estimate $\mathbb{E}_{\theta \sim s_{d-1}}[f(\theta)]$. Writing $\theta_1, \ldots, \theta_M$ i.i.d. samples of $\theta \sim s_{d-1}$ and $Y_1, \ldots, Y_M$ $M$ independent copies of the random centered vector $Y$, we consider the following estimator

$$\frac{1}{M} \sum_{i=1}^{M} [f(\theta_i) - \beta^T Y_i],$$

where $\beta \in \mathbb{R}^p$ is a constant vector to be determined. The variance of this estimator is proportional to $\mathrm{Var}(f(\theta) - \beta^T Y)$. It follows that if we write $\beta^*$ the parameter minimizing this variance, then the pair $(\mathbb{E}(f(\theta)), \beta^*)$ is solution of the least square problem

$$\min_{(\zeta, \beta) \in \mathbb{R} \times \mathbb{R}^p} \mathbb{E}[(f(\theta) - \zeta - \beta^T Y)^2].$$

An empirical version of this quadratic problem on a sample $(\theta_1, \ldots, \theta_M)$ writes

$$(\widehat{\mathbb{E}(f(\theta))}_M, \beta_M) = \arg\min_{\zeta, \beta \in \mathbb{R} \times \mathbb{R}^p} \|\mathbf{F} - \zeta \mathbb{1}_M - \mathbf{Y}\beta\|_2^2 \tag{15}$$

where $\mathbf{F} = \big(f(\theta_i)\big)_{i=1,\ldots,M}^T$, $\mathbb{1}_M = (1, \ldots, 1)^T \in \mathbb{R}^M$, and $\mathbf{Y} = \big(Y_i^T\big)_{i=1,\ldots,M} \in \mathbb{R}^{M \times p}$.

Recently, Nguyen & Ho [2024] introduced a Sliced Wasserstein distance estimation using Gaussian control variates and Leluc et al. [2024] developped a method using spherical harmonics control variates. We focus only on Leluc et al. [2024] here, since their method yields much better experimental results. In their work, Leluc et al. [2024] chose Spherical Harmonics [Müller, 1998] as control variates. Spherical harmonics are functions which form an orthonormal basis $(\phi_i)$ of the Hilbert space $L^2(\mathbb{S}^{d-1}, s_{d-1})$. In this setting, the random variable $Y$ is thus chosen as $Y = (\phi_i(\theta))_{i=1,\ldots,p}$, with $\theta \sim s_{d-1}$. In practice, the number $p$ is chosen as $p = L_{n,d} = \sum_{l=1}^{n} N(d, 2l)$, the number of spherical harmonics of even degree up to $2n$, with $N(d, n) = (2n + d - 2)\frac{(n+d-3)!}{(d-2)!n!}$ the number of spherical harmonics of degree $n$ in dimension $d$.

Leluc et al. [2024] then compute the solution $(SHCV_{M,n}^2, \beta_M)$ of (15) on a sample $(\theta_1, \ldots, \theta_M)$ and use the control variates estimator $SHCV_{M,n}^2$ as estimator of the (squared) Sliced Wasserstein distance.

They prove the following convergence property.

**Proposition 6 :** Let $\mu, \nu$ be two discrete measures in $\mathbb{R}^d$ with finite moments of order 2 and let $d \geq 2$. For any sequence of degrees $n = (n_M)_M$ such that $n_M = o\big(M^{1/(2(d-1))}\big)$ as $M \longrightarrow +\infty$, we have

$$\big|SHCV_{M,n}^2(\mu, \nu) - SW_2^2(\mu, \nu)\big| = \mathcal{O}_{\mathbb{P}}\left(\frac{1}{nM^{1/2}}\right), \tag{16}$$

where the notation $X_n = \mathcal{O}_{\mathbb{P}}(a_n)$ means that the sequence $\frac{X_n}{a_n}$ is stochastically bounded [3].

Notice that since $n_M = o\big(M^{1/(2(d-1))}\big)$, in high dimensions $d$ the global convergence rate is similar to that of the classical Monte Carlo method described in Sec. 3.1.1.

# 4 Experimental results

This section presents experimental results from all the different sampling strategies presented in Sec. 3, on a variety of datasets. To provide representative results, we select datasets spanning a range of dimensions going from 2 to $28 \times 28$. Those include a toy dataset and three "real-life" ones. We first present results on Gaussian mixtures in the following dimensions $\{2, 3, 5, 10, 20, 50\}$, the six ground truths (true distances) are estimated using $10^8$ projections. Secondly, we show some dimensionality reduction results on 12 different datasets of persistence diagrams (for the case of 2 dimensional discrete measures). Then we show some convergence results in the specific case of measures in 3 dimensions. Specifically, we compare different estimations of the Sliced Wasserstein distance between 3D point clouds taken from the ShapeNetCore dataset, see [Chang et al., 2015]. Finally we compare different dimensionality reduction results on the MNIST dataset [LeCun, 1998]. For the experiments on the Gaussian mixtures we compare the listed strategies with the following sampling

---

[3]The notation $X_n = \mathcal{O}_{\mathbb{P}}(a_n)$ means that for all $\epsilon > 0$, there exists finite $K > 0$ and $N > 0$ such that $\mathbb{P}[|X_n| > Ka_n] < \epsilon$ for all $n > N$.

numbers $\{100, 300, 500, 700, 900, 1100, 2100, 3100, 4100, 5100, 6100, 7100, 8100, 9100, 10100\}$. Otherwise, we use the following sampling numbers $\{100, 1100, 2100, 3100, 4100, 5100, 6100, 7100, 8100, 9100, 10100\}$. Tab. 3 displays the acronyms of all the sampling methods compared in the following experiments. For each sampling method from Tab. 3, there are two variants finishing with the term "Area Mapped" and two variants finishing with the term "Normalized Mapped". The first one means that we applied the equal area mapping detailed in paragraph 3.2.1.4. The second one means we normalize each point generated by those methods, this normalization method is also detailed in paragraph 3.2.1.4.

| Name | Legends | Dimensions |
|---|---|---|
| Riesz Randomized | R.R. | 2,3,5,10,20,50 |
| Uniform Sampling | U.S. | 2,3,5,10,20,50 |
| Othornormal Sampling | O.S. | 2,3,5,10,20,50 |
| Halton Area Mapped | H.A.M. | 2,3 |
| Halton Randomized Area Mapped | H.R.A.M. | 3 |
| Halton Normalized Mapped | H.N.M. | 5,10,20,50 |
| Halton Randomized Normalized Mapped | H.R.N.M | 5,10,20,50 |
| Fibonacci Point Set | F.P.S. | 3 |
| Fibonacci Randomized Point Set | F.R.P.S. | 3 |
| Sobol Area Mapped | S.A.M. | 3 |
| Sobol Randomized Area Mapped | S.R.A.M. | 3 |
| Sobol Normalized Mapped | S.N.M. | 5,10,20 |
| Sobol Randomized Normalized Mapped | S.R.A.M. | 5,10,20 |
| Spherical Harmonics Control Variates | S.H.C.V. | 3,5,10,20 |
| Spherical Sliced Wasserstein Randomized | S.S.W.R. | 3,5,10,20,50 |

Table 3: For each method used in this experimental part, associated acronym, and list of dimensions where this method is used.

## 4.1 Implementation of the sampling methods

This section provides details on the implementations used for the sampling methods, and specifies how the parameters are set. The implementations used are grouped and are available here https://github.com/Keanu-Sisouk/SW-Sampling-Guide.

- **Classical M.C. methods:** For both methods we used python included functions to sample a Gaussian variable and to sample orthogonal matrices in $d$ dimension. For sampling orthogonal matrices we use the following python library scipy.stats.ortho_group https://docs.scipy.org/doc/scipy/reference/generated/scipy.stats.ortho_group.html.

- **Halton & Sobol sequences:** In dimension 3 and less, we use python implementations from the library scipy.stats.qmc (https://docs.scipy.org/doc/scipy/reference/generated/scipy.stats.qmc.Halton.html & https://docs.scipy.org/doc/scipy/reference/generated/scipy.stats.qmc.Sobol.html. As for the parameters we set "scramble" to True to get the randomized version. For high dimensions, we use Leluc et al. [2024]'s implementation available here https://github.com/RemiLELUC/SHCV.

  **Remark 8 :** For the Sobol sequence, we noticed that the implementation provided by Leluc et al. [2024] cannot be used in dimension higher than 20.

- **Riesz point configuration:** We use a code provided by François Clement (https://sites.math.washington.edu/~fclement/), implementing a projected gradient descent method, where we choose the number of iterations as $T = 10$, the gradient step as 1 and $s = 0.1$. The function can be found in the riesz_noblur.py script in the repository https://github.com/Keanu-Sisouk/SW-Sampling-Guide.

- **Spherical Sliced Wasserstein:** We used the following implementation from Bonet et al. [2023] that can be found in POT library (Python Optimal Transport) https://pythonot.github.io/

`auto_examples/backends/plot_ssw_unif_torch.html`. For the hyper-parameters we set the number of iteration $T = 250$, the learning rate $\epsilon = 150$ and the number of great circles $L = 500$. For the initialization, we generate $\theta_1, \ldots, \theta_M \sim s_{d-1}$ following the method described in Sec. 3.1.1.

- **Spherical Harmonics Control Variates:** We use the implementation provided by Leluc et al. [2024], available in `https://github.com/RemiLELUC/SHCV`. They provide two possible functions **SHCV** and **SW_CV**, both functions return a value of a SW estimate. These functions differ in the way they implement the optimization of Eq. 15. Depending on the number of control variates, one of the functions is much more efficient than the other. For this reason, in our experiments, we use both functions and always keep only the minimal error among the two.

## 4.2 Gaussian data

This part details the experiments on a toy dataset chosen because it is simple to replicate and simple to understand. We compare different estimates of $SW_2^2(\mu_d, \nu_d)$ for $d \in \{2, 3, 5, 10, 20, 50\}$. We pick up Leluc et al. [2024]'s setting, using $\mu_d = \frac{1}{N} \sum_{i=1}^{N} \delta_{x_i}$ and $\nu_d = \frac{1}{N} \sum_{i=1}^{N} \delta_{y_i}$ with $x_1, \ldots, x_N \sim \mathcal{N}(x, \mathbf{X})$, $y_1, \ldots, y_N \sim \mathcal{N}(y, \mathbf{Y})$, where $N = 1000$. The means are drawn as $x, y \sim \mathcal{N}(\mathbb{1}_d, I_d)$ and the covariances are $X, Y = \Sigma_x \Sigma_x^T, \Sigma_y \Sigma_y^T$ where all entries of the matrices are drawn using the standard normal distribution. In Fig. 4.1, we show convergence curves generated by all the different sampling strategies in all the dimensions listed above. Fig. 4.2 reports the distance estimation error as a function of computation time (in seconds). In both figures, both axes are log scaled. We can see in Fig. 4.1 that up to dimension 5, Q.M.C. methods are preferable convergence wise, then the orthonormal sampling is preferable in dimension 20 and 50. In contrast, we can see in Fig. 4.2 that for dimensions less than 10, the S.H.C.V. method has a better error, with similar running time. For higher dimensions, however, the orthonormal sampling is much faster, for a given error target.

**Remark 9 :** One may notice in Fig. 4.1b that both the S.H.C.V. method and the Q.M.C. method with the s-Riesz points (R.R.) reach a plateau at around $10^3$ projections. Our hypothesis is that both methods have a better estimation of $SW_2^2$ than the simple random sampling with $10^8$ projections that we use as a ground truth. We test this hypothesis in a simple case where $SW_2^2(\mu, \nu)$ can be computed explicitly. We define $\mu = \frac{1}{2}[\delta_{x_1} + \delta_{x_2}]$ and $\nu = \frac{1}{2}[\delta_{y_1} + \delta_{y_2}]$, with $x_1, x_2 = (1, 0, 0)^T, (0, -1, 0)^T$ and $y_1, y_2 = (0, 0, 1)^T, (0, 0, -1)^T$. Simple computations yield $SW_2^2(\mu, \nu) = \frac{2(\pi - \sqrt{2})}{3\pi}$. Knowing the true value of $SW_2^2(\mu, \nu)$, we find that with $10^4$ points, the Q.M.C. method with the s-Riesz points configuration and the S.H.C.V. methods already have errors one order smaller that ones made by uniform sampling with $10^8$ points.

**Remark 10 :** Note that for the running time curves, we do not include the s-Riesz points configuration starting from the dimension 3 because it takes around $10^2$ seconds to generate $10^3$ points and $9 \times 10^3$ seconds to generate $10^4$ points. However, observe that those points, once generated, can be stored once for all to compute other $SW_2^2$ distances or any other Monte Carlo estimation problems for functions defined on the unit sphere. This means that these configurations should not be discarded by default. For practical applications where the number of $SW_2^2$ distances to compute is large, the computing time for these configurations can be factorized by the number of distances to compute and hence could become a negligible factor when the sampling number is moderate.

**Remark 11 :** Recalling the running time complexity $\mathcal{O}(TM^2)$ in paragraph 3.2.2.2 and the running time results above, this shows that one needs to spend $9 \times 10^7$ seconds to generate $10^6$ points. This demonstrates the limitation of this sampling method in terms of scalability, in other words when one needs a very large sampling number.

## 4.3 Persistence diagrams reduction dimension score

The goal of this section is to evaluate the relevance of the sampling methods studied in Sec. 3, in the context of a concrete use case, involving two-dimensional real-life datasets. For that, we focus in this section on persistence diagrams, a popular object used in Topological Data Analysis [Edelsbrunner & Harer, 2009]. Persistence diagrams are data abstractions encapsulating the features of interest of complex input datasets

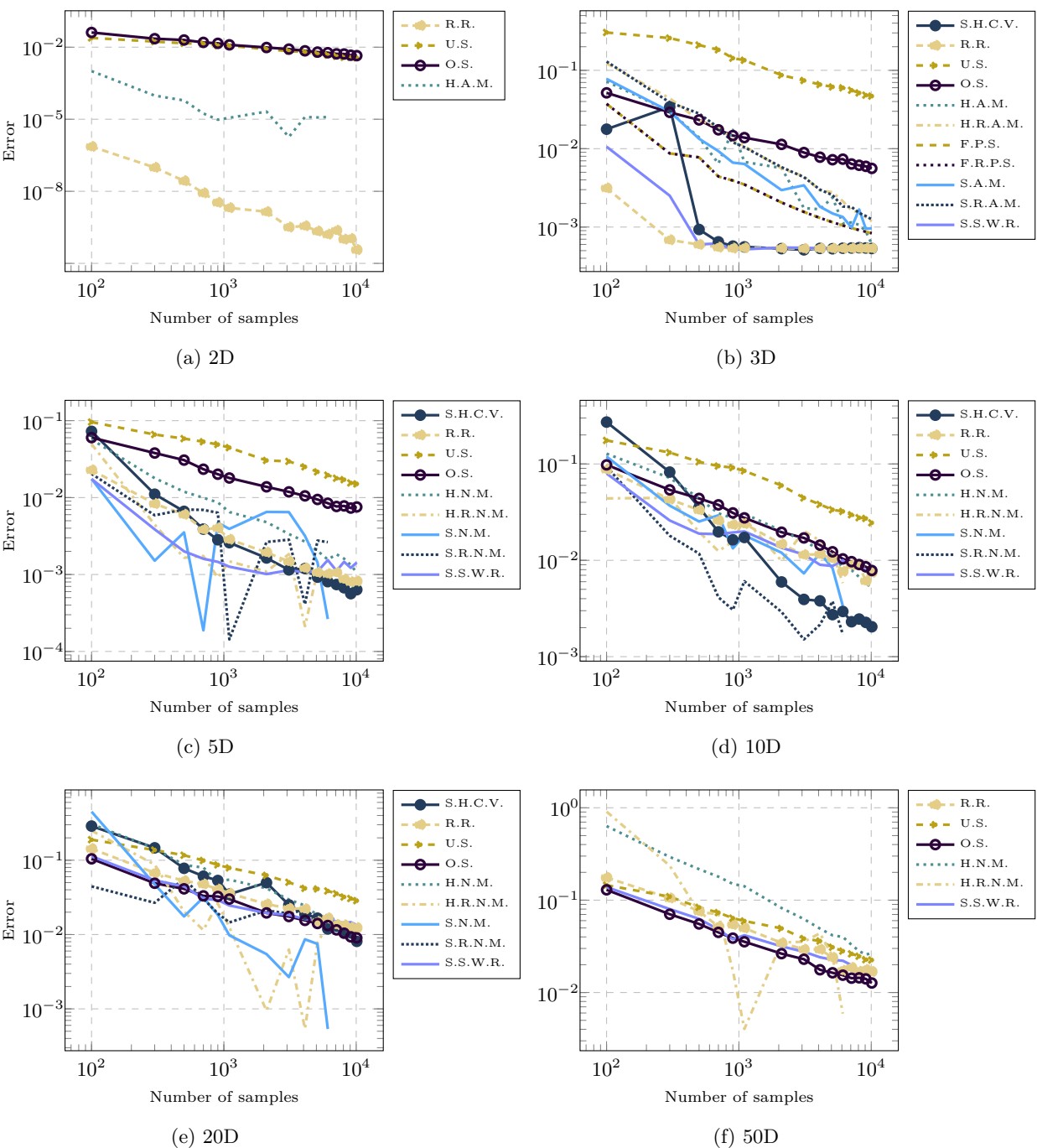

Figure 4.1: Comparison of convergence rate results for the studied sampling methods (Gaussian data, Sec. 4.2).

(e.g. scalar fields) into simple two-dimensional representations. Specifically, we consider an input dataset represented as a piecewise linear (PL) scalar field, namely a function $f : \mathcal{M} \to \mathbb{R}$ defined on a PL $(d_{\mathcal{M}})$-manifold $\mathcal{M}$ with $d_{\mathcal{M}} \leq 3$. Take a value $a \in \mathbb{R}$, we denote $f_{-\infty}^{-1}(a) = f^{-1}(] - \infty, a])$ the sub-level set of $f$ at $a$. While increasing $a$, the topology of $f_{-\infty}^{-1}(a)$ changes at the critical points of $f$ in $\mathcal{M}$. Those critical points are classified by their index $\mathcal{I}$: 0 for minima, 1 for 1-saddles, $d_{\mathcal{M}} - 1$ for $(d_{\mathcal{M}} - 1)$-saddles and $d_{\mathcal{M}}$ for maxima. Following the Elder rule [Edelsbrunner & Harer, 2009], a topological feature of $f_{-\infty}^{-1}(a)$ (connected component, cycle, void) is associated with a pair of critical points $(c, c')$ such that $f(c) < f(c')$

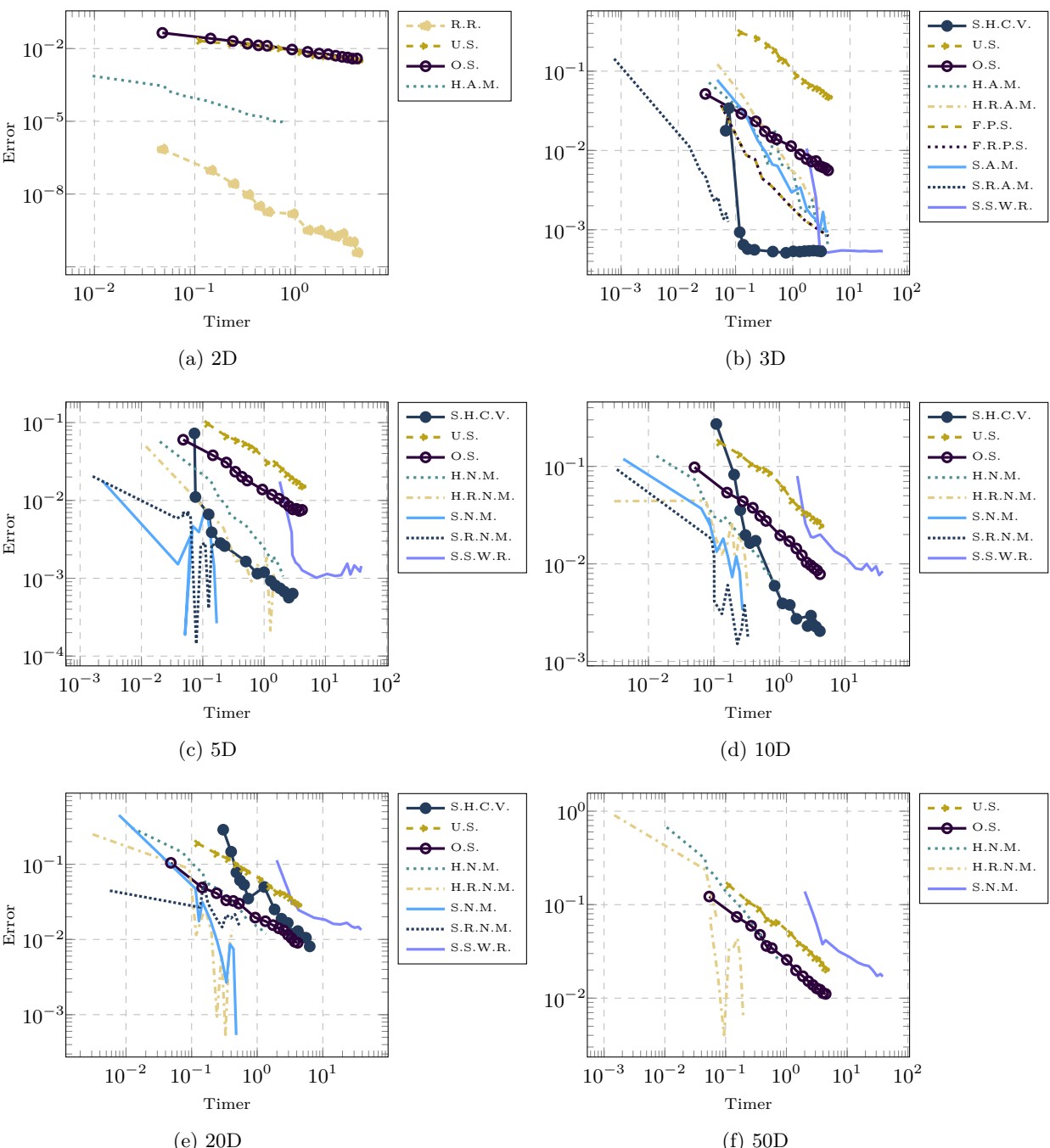

Figure 4.2: Distance estimation error as a function of computation time (seconds). Computation times include the point generation as well as the $SW_2^2$ distance approximation.

and $\mathcal{I}_c = \mathcal{I}_{c'} - 1$. This pair corresponds to the *birth* and *death* of the topological feature during the sweep of the range from $-\infty$ to $+\infty$ by $a$, and it is called a *persistence pair*. As an example, when two connected components of $f_{-\infty}^{-1}(a)$ merge at a critical point $c'$, the younger one (created last) *dies* to let the older one (created first) live on. Then those persistence pairs are represented as 2D points where the horizontal coordinate corresponds to the *birth* of a topological feature (noted $b = f(c)$) and where the vertical one corresponds to its *death* (noted $d = f(c')$). The lifespan of a feature is called *persistence* and is simply encoded as $b - d$. This representation is called the *Persistence Diagram*, and its popularity in topological

data analysis is explained by its stability to the addition of noise. See Fig. 4.3 for a simple example of a persistence diagram.

**Remark 12 :** Two persistence diagrams can have a different number of points, so to make it a balanced transport problem one has to augment them. Formally, denoting $d_1 = \frac{1}{N_1} \sum_{k=1}^{N_1} \delta_{x_k}$ , $d_2 = \frac{1}{N_2} \sum_{k=1}^{N_2} \delta_{y_k}$ the diagrams, and noting $\Delta_{d_1} = \frac{1}{N_1} \sum_{k=1}^{N_1} \delta_{\pi_\Delta(x_k)}$, $\Delta_{d_2} = \frac{1}{N_2} \sum_{k=1}^{N_1} \delta_{\pi_\Delta(y_k)}$ their projections on the diagonal $\Delta$, one considers $\mu = \frac{1}{N}[N_1 d_1 + N_2 \Delta_{d_2}]$ and $\nu = \frac{1}{N}[N_2 d_2 + N_1 \Delta_{d_1}]$ as input measures with $N = N_1 + N_2$. Then the Sliced Wasserstein distance can be used to compare persistence diagrams as detailed by Carrière et al. [2017].

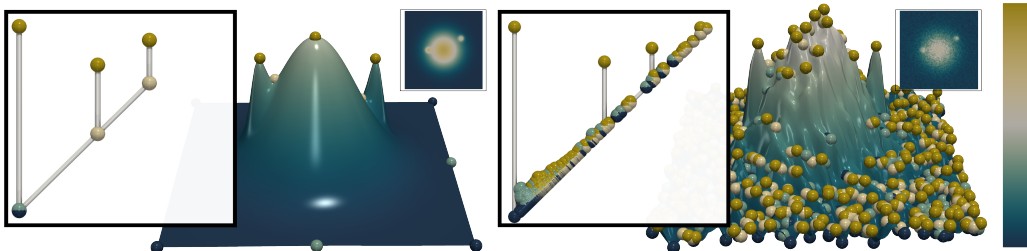

Figure 4.3: A simple example of a persistence diagram issued from a gaussian mixture (left). On the right you can see that the persistence diagram is stable to the addition of noise.

We present dimensionality reduction results on 12 ensembles of persistence diagrams [Pont et al., 2021] described in [Pont et al., 2022], which original scalar fields include simulated and acquired 2D and 3D ensembles from SciVis constests [Organizers, 2004]. The dimensionality reduction techniques used are MDS [Kruskal & Wish, 1978] and t-SNE [van der Maaten & Hinton, 2008] applied on distance matrices obtained by the SW estimations between the persistence diagrams. For a given technique, one quantifies its ability to preserve the cluster structure of an ensemble by running the *k*-means algorithm in the resulting 2D-layouts. Then one evaluates the quality of the clustering with the normalized mutual information (NMI) and adjusted rand index (ARI), which should both be equal to 1 for a clustering that is identical to the classification ground-truth. Tab. 4 shows the average clustering scores of both MDS [Kruskal & Wish, 1978] and t-SNE [van der Maaten & Hinton, 2008]. First we take the average from distance matrices made by each $SW_2^2$ estimates on all sampling number $\{100, 1100, 2100, 3100, 4100, 5100, 6100, 7100, 8100, 9100, 10100\}$. Then we average again over all the 12 different ensembles of persistence diagrams. One can see that all the methods are quite similar. But overall the s-Riesz points configuration, which are just the *M*-th unity roots up to a rotation, is slightly better.

Table 4: Average NMI and ARI scores for over all 12 ensembles of persistence diagrams.

| Method | MDS NMI | t-SNE NMI |
|---|---|---|
| Riesz | 0.74 | 0.65 |
| Uniform | 0.74 | 0.59 |
| Orthonormal | 0.75 | 0.63 |
| Halton | 0.74 | 0.58 |

| Method | MDS ARI | t-SNE ARI |
|---|---|---|
| Riesz | 0.64 | 0.51 |
| Uniform | 0.64 | 0.44 |
| Orthonormal | 0.64 | 0.48 |
| Halton | 0.63 | 0.41 |

### 4.4 3D Shapenet 55Core Data

This part details convergence results on a 3D dataset commonly used as a benchmark when studying shape comparison techniques. So as in [Nguyen et al., 2024] and [Leluc et al., 2024], we took three 3D point clouds issued from the ShapenetCore dataset introduced by [Chang et al., 2015]. Among the different shapes in the dataset, we took one lamp, one plane and one bed; with all three of them having $N = 2048$ points. Fig. 4.4 displays the three datasets considered for this experiment.

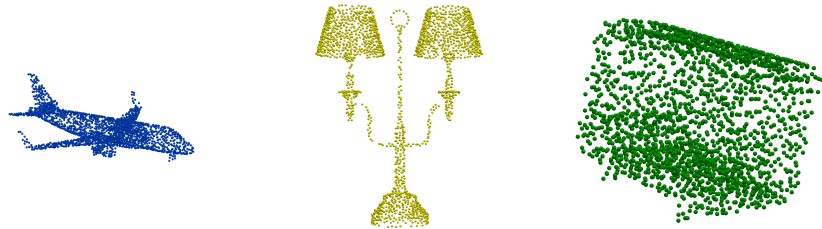

Figure 4.4: The three point clouds taken from the ShapenetCore dataset (a plane, a lamp and a bed).

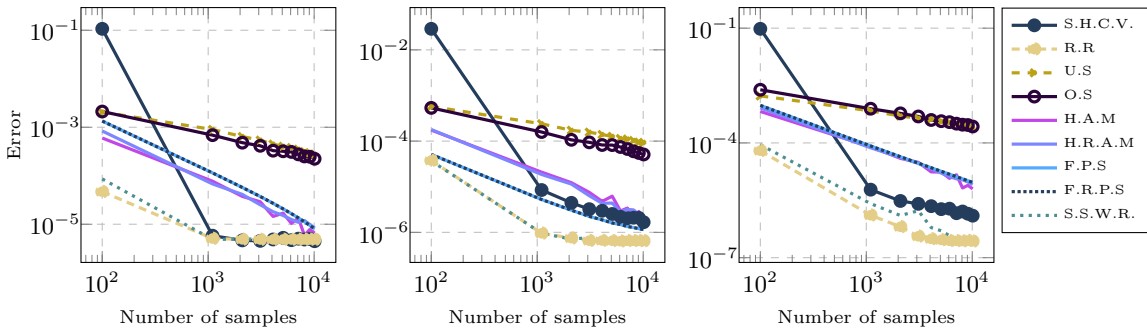

Figure 4.5: Comparison of convergence rate results from the different sampling methods. The first plot shows errors made with respect to the $SW_2^2$ distance between a lamp and a plane. The second one is between a plane and a bed. The last one corresponds to $SW_2^2$ between a plane and a bed.

Fig. 4.5 shows different convergence curves of Sliced Wasserstein estimates between the three point clouds. As in Sec. 4.2, the methods dominating are the Q.M.C., R.Q.M.C., S.S.W. and S.H.C.V. methods, especially the s-Riesz points configuration and the Spherical Sliced Wasserstein sampling.

### 4.5 MNIST reduction dimension score

The goal of this section is twofold. First, it evaluates the practical convergence of the studied sampling methods on real-life high-dimensional datasets. Second, it describes an application of the SW distance for high-dimensional data, namely, dimensionality reduction. For this, we select the classical MNIST dataset [LeCun, 1998]. To construct our dataset, we represent each digit image as a point in $\mathbb{R}^{28 \times 28}$. For each class $\{0, 1, 2, 3, 4, 5, 6, 7, 8, 9\}$, we select randomly 600 digit images and divide them into groups of 200. This results in 30 point clouds of 200 points each, in $\mathbb{R}^{28 \times 28}$, with 10 ground-truth classes. Fig. 4.6 illustrates the $30 \times 30$ matrix of $SW$ distances between all point clouds in the dataset. We use MDS and t-SNE to produce 2D layouts from the distance matrices generated by the various sampling methods with different sample sizes. We then apply a clustering algorithm to these 2D layouts and average the clustering scores (NMI and ARI, see Sec. 4.3) on all sampling numbers for all the studied sampling strategies. Results are provided in Tab. 5. In such high dimension ($d = 784$), we see that the performance of L.D.S. collapse, the three sampling methods standing out being the s-Riesz points configuration, the uniform sampling and the orthonormal sampling.

## 5 Recommendation & conclusion

In this paper, we have studied several sampling strategies on the sphere for computing an approximation of the Sliced Wasserstein distance.

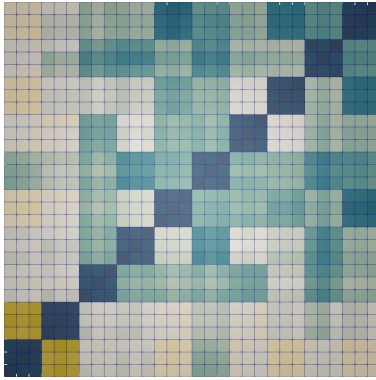

Figure 4.6: Sliced Wasserstein distance matrix of our dataset using $10^6$ projections. All 10 classes, $\{0, 1, 2, 3, 4, 5, 6, 7, 8, 9\}$, of 3 members each are well represented in the matrix.

Table 5: Average NMI and ARI scores with standard deviation. Higher scores correspond to better clustering.

| Method | MDS NMI | t-SNE NMI |
|---|---|---|
| Riesz | $1 \pm 0.$ | $0.98 \pm 2e\text{-}2$ |
| Uniform | $1 \pm 0.$ | $0.97 \pm 4e\text{-}2$ |
| Orthonormal | $1 \pm 0.$ | $0.98 \pm 3e\text{-}2$ |
| Halton | $0.91 \pm 1e\text{-}1$ | $0.91 \pm 9e\text{-}2$ |
| S.S.W. | $1 \pm 0.$ | $0.98 \pm 4e\text{-}2$ |

| Method | MDS ARI | t-SNE ARI |
|---|---|---|
| Riesz | $1 \pm 0.$ | $0.95 \pm 7e\text{-}2$ |
| Uniform | $1 \pm 0.$ | $0.91 \pm 1e\text{-}1$ |
| Orthonormal | $1 \pm 0.$ | $0.94 \pm 8e\text{-}2$ |
| Halton | $0.75 \pm 2e\text{-}1$ | $0.76 \pm 2e\text{-}1$ |
| S.S.W. | $1 \pm 0.$ | $0.94 \pm 1e\text{-}1$ |

Regarding theoretical guarantees, this study highlighted the following limitations. The classical i.i.d. sampling benefits from theoretical guarantees with a convergence rate in $O(1/\sqrt{M})$ and a time complexity linear in the number $M$ of projections. Orthonormal sampling and L.D.S. such as Halton or Sobol lack convergence rate guarantees on the sphere (these guarantees being only obtained for sequences on hypercubes for L.D.S). As for deterministic point generation methods (like Riesz), the Sliced Wasserstein integrand also lacks sufficient regularity to guarantee results in dimensions higher than 2.

While lacking theoretical guarantees in terms of convergence, the experimental study suggests that Q.M.C methods (L.D.S. or s-Riesz points) provide competitive results in small to intermediate dimension, while having a similar convergence rate to classical random sampling methods in intermediate to higher (for Riesz) dimensions. These results seem to indicate that, while $f$ is not regular enough for the convergence guarantees detailed in this paper, there may be some non-proven convergence results requiring weaker regularity conditions that would be applicable to $SW$.

Now, considering computation times, as shown by Fig. 4.2 and Tab. 2, classical i.i.d. sampling remains the slowest method in all our experiments. While orthonormal sampling lacks theoretical guarantees, it seems to be one of the most efficient methods whatever the dimension, and is particularly competitive in high dimensions, with a very reasonable increase of computation time. L.D.S. methods also remain competitive in pratice for small dimensions. s-Riesz points, while competitive in terms of convergence rate, have a prohibitive time complexity in $O(M^2)$, which makes them completely unsuitable for a large number of projections.

The experiments also suggest that the S.H.C.V. method is very competitive in intermediate dimensions, while becoming less efficient when $d$ increases.

Based on the different experimental results provided in this paper, we make the following recommendations:

- For small dimensions (less than 3), Q.M.C. methods such as s-Riesz points or L.D.S. mapped onto the sphere can be privileged with respect to uniform sampling,

- For high dimensions (greater than 20), the orthonormal sampling method emerges as the most suitable choice. It is also one of the simplest methods to implement, which makes it particularly attractive in practice.

- For intermediate dimensions (between 5 and 10), choosing an appropriate method should depend on the experimental requirements. Spherical harmonics are an excellent option if computational resources are limited and if the number of $SW$ distances to be computed is low. However, it is worth noting that some Q.M.C. strategies, being independent of the input measures, have the advantage of allowing the generated points to be reused and of allowing an independent computation in $M$ (except the Riesz points). This should be particularly beneficial when a high number of projections is required and a large number of $SW$ distances must be computed. In such cases, we suggest to store the samples to factorize the computing time across experiments.

## Acknowledgments

This work is partially supported by the European Commission grant ERC-2019-COG "TORI" (ref. 863464, https://erc-tori.github.io/). JD acknowledge the support of the "France 2030" funding ANR-23-PEIA-0004 ("PDE-AI") and the support of the funding SOCOT - ANR-23-CE40-0017.

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
