# OpenReview forum: "A User's Guide to Sampling Strategies for Sliced Optimal Transport"
_TMLR — Accepted by TMLR_

### Review · Reviewer_b6ih · 2025-03-10

**Summary Of Contributions:**

This paper studies the approximation of the Sliced-Wasserstein distance by comparing different approximations of the integral with respect to the uniform measure on the sphere. It begins by providing a theoretical analysis of the regularity of the integrand of the Sliced-Wasserstein distance. Then, it describes different strategies to estimate the integral, and provides complexity and known theoretical results for the different strategies. Finally, the authors provide a comparison of the described strategies in different settings, and make recommendations on which method to use given the setting.

**Audience:**

Yes

**Claims And Evidence:**

Yes

**Requested Changes:**

As there are a lot of results, it would be nice in addition to Table 1 to also summarize in a Table the known results of the different method (i.e. the rates, the computational complexities...).

For the timings reported in Figure 4.2, I am wondering if there are all optimal. I had a quick look at the code, and it seems that the uniform sampling normalization is done with a loop, while it could be done in parallel (which might speed things up a bit?). I also think that the orthonormal sampling could be done in parallel by first sampling a $(M\times d\times d)$ matrix with independent Gaussian components, and then taking the QR decomposition, see e.g. [1] or [2]. I don't know if it would do a big difference though.

I am also wondering if sampling uniform points on the sphere by e.g., minimizing the spherical Sliced-Wasserstein distance using the closed-form in Proposition 1 of [2], could provide good results?


Typos:
- Page 2: "Indeed the higher the dimension the harder it is to generate samplings with Q.M.C. and R.Q.M.C.
approaches." -> to generated samples?
- Equation 11: "$\mathbb{E}_{\theta\sim \mathcal{U}(\mathbb{S}^{d-1})]}$"
- In the proof of Proposition 1, it is written "On each $V_l$, $f_{|\mathbb{S}^{d-1}}=f_{\sigma_l \mathbb{S}^{d-1}}$. I guess it should be $=f_{\sigma_l}$?
- Table 1: "Incremental computation": I am not sure to understand what it means. I would say that incremental computations are when we can't compute points in parallel.
- Section 3.1.2 title: "Orthonornal"
- Page 13: "s-Riesz energy for $s=-1$", but is only defined for $s\ge 0$?
- 2 citations of the same paper "Sliced Wasserstein Auto-encoders" (with publication dates 2018 and 2019)


[1] Lin, T., Zheng, Z., Chen, E., Cuturi, M., & Jordan, M. I. (2021, March). On projection robust optimal transport: Sample complexity and model misspecification. In International Conference on Artificial Intelligence and Statistics (pp. 262-270). PMLR.

[2] Bonet, C., Berg, P., Courty, N., Septier, F., Drumetz, L., & Pham, M. T. (2022). Spherical sliced-wasserstein. arXiv preprint arXiv:2206.08780.

**Strengths And Weaknesses:**

This work provides a nice overview of different strategies to approximate the Sliced-Wasserstein distance with Monte-Carlo estimators. In addition to compare known methods (such as uniform sampling, orthonormal sampling, and some of the QMC methods), it also compares other methods which have not been tested to compute SW to the best of my knowledge (e.g. Fibonacci point sets, or minimizers of Riesz energy).

I believe this is a valuable contribution to the community, as it provides a useful summary for practitioners on selecting an estimate Sliced-Wasserstein distance.

**Strengths:**
- Analysis of the regularity of the integrand
- Clear description of different strategies to approximate integrals with respect to the uniform measure on the sphere, with known results on the theoretical side
- Comparison of the methods in several settings
- Guidelines on which method to choose given the setting


**Weaknesses:**
- As acknowledged in the paper, some methods lack theoretical guarantees on the sphere
- One might say that MNIST is not a very high dimensional setting, and it could be interesting to see what happens on an even higher dimensional dataset.

---

> ### Author Response · Authors · 2025-03-30
> **Changes done on submission for reviewer 1**
>
> First, thank you for your insightful feedback. Taking your comments into account has allowed us to improve the quality of our work.
>
> -As there are a lot of results, it would be nice in addition to Table 1 to also summarize in a Table the known results of the different methods (i.e the rates, the computational complexities…)
>
> Thank you for this suggestion. We added a table 2 indicating the rates and computational complexities.
>
> -For the timings reported in Figure 4.2, I am wondering if there are all optimal. I had a quick look at the code, and it seems that the uniform sampling normalization is done with a loop, while it could be done in parallel (which might speed things up a bit?). I also think that the orthonormal sampling could be done in parallel by first sampling a (M x d x d) matrix with independent Gaussian components, and then taking the QR decomposition, see e.g [1] or [2]. I don’t know if it would do a big difference though.
>
> We have already considered doing uniform sampling in parallel, however there are two reasons why we didn’t keep this approach for implementations.
>
> First, uniform sampling is done by generating and normalizing Gaussian vectors. When such a vector has a very small norm, the normalization step causes instability, and this instabilities had an impact on the SW computation, so we decided to discard such vectors, which is easier with a loop.
>
> Besides, for the number of samplings shown in the paper, the real bottleneck for the computation time lies in the computation of the Sliced Wasserstein approximation. Meaning that the computation time of point generation is “negligible”, the gain obtained by using parallel sampling (for uniform or orthogonal sampling) is negligible.
>
> - I am also wondering if sampling uniform points on the sphere by e.g., minimizing the spherical Sliced-Wasserstin distance using the closed-form in Proposition 1 of [2], could provide good results?
>
> To our understanding, minimizing (13) in Proposition 1 of [2] leads to getting a uniform grid on $\mathbb{S}^1$ (i.e the i/N where N is the number of points). We don’t see how we could extrapolate that to the hypersphere of higher dimensions?
>
> - "Incremental computation''
>
> By incremental computation, we mean that each sampling can be computed independently from the others. This means that when one wants to generate M samplings after having already generated N samplings (N < M), one only needs to generate M-N+1 samplings instead of regenerating all the M samplings starting from zero.
>
> - Table showing convergence rate and time complexity
>
> We have added the table after the table 1 page 7, we have also added the space complexity.

---

> > ### Comment · Reviewer_b6ih · 2025-03-31
> >
> > Thank you for your answer and addressing my comments. I still have some comments.
> >
> > **First, uniform sampling is done by generating and normalizing Gaussian vectors. When such a vector has a very small norm, the normalization step causes instability, and this instabilities had an impact on the SW computation, so we decided to discard such vectors, which is easier with a loop.**
> >
> > Sure, instabilities could happen. I don't think I experimented such issues though (and the implementation in POT https://github.com/PythonOT/POT/blob/master/ot/sliced.py does not seem either to take these into accounts).
> >
> >
> > **Besides, for the number of samplings shown in the paper, the real bottleneck for the computation time lies in the computation of the Sliced Wasserstein approximation. Meaning that the computation time of point generation is “negligible”, the gain obtained by using parallel sampling (for uniform or orthogonal sampling) is negligible.**
> >
> > I agree that the runtime of this part should be negligible, it was just a suggestion to remove the loop which might slightly change the results of Figure 4.2.
> >
> >
> > **To our understanding, minimizing (13) in Proposition 1 of [2] leads to getting a uniform grid on $\mathbb{S}^1$ (i.e the i/N where N is the number of points). We don’t see how we could extrapolate that to the hypersphere of higher dimensions?**
> >
> > The projection of the uniform measure on the sphere $S^{d-1}$ to $S^1$ is the uniform measure on $S^1$ (see e.g. Appendix B.3 in [2]). Thus, you could minimize SSW with the closed form of $W_2$ on $S^1$ to generate samples from the uniform measure on the sphere. This is actually implemented in POT as ot.sliced_wasserstein_sphere_unif (see e.g. https://pythonot.github.io/auto_examples/backends/plot_ssw_unif_torch.html#sphx-glr-auto-examples-backends-plot-ssw-unif-torch-py for an example of its minimization).
> >
> > I do not expect it to work perfectly, but an additional comparison with this method could be nice for a more exhaustive set of comparisons. However, this is not crucial, and I will recommend acceptance anyway.
> >
> >
> > **By incremental computation, we mean that each sampling can be computed independently from the others. This means that when one wants to generate M samplings after having already generated N samplings (N < M), one only needs to generate M-N+1 samplings instead of regenerating all the M samplings starting from zero.**
> >
> > I understand, but maybe there would be a less ambiguous word for it? e.g. Independent sampling.

---

> > > ### Author Response · Authors · 2025-04-01
> > > **Answer to reviewer 1**
> > >
> > > Thank you for the fast answers and clarifications.
> > >
> > > - Sure, instabilities could happen. I don't think I experimented such issues though (and the implementation in POT https://github.com/PythonOT/POT/blob/master/ot/sliced.py does not seem either to take these into accounts). I agree that the runtime of this part should be negligible, it was just a suggestion to remove the loop which might slightly change the results of Figure 4.2.
> > >
> > > Of course we will add those versions of implementation accordingly as an alternative in our repository. With the experiments we ran we did not see noticeable changes in the running time results for those number of samplings mentioned in the paper at the very least (not taking account the instabilities we noticed for the uniform sampling).
> > >
> > > - The projection of the uniform measure on the sphere $\mathbb{S}^{d-1}$ to $\mathbb{S}^1$ is the uniform measure on (see e.g. Appendix B.3 in [2]). Thus, you could minimize SSW with the closed form of $W_2$ on $\mathbb{S}^1$ to generate samples from the uniform measure on the sphere. This is actually implemented in POT as ot.sliced_wasserstein_sphere_unif (see e.g. https://pythonot.github.io/auto_examples/backends/plot_ssw_unif_torch.html#sphx-glr-auto-examples-backends-plot-ssw-unif-torch-py for an example of its minimization). I do not expect it to work perfectly, but an additional comparison with this method could be nice for a more exhaustive set of comparisons. However, this is not crucial, and I will recommend acceptance anyway.
> > >
> > > Thank you for the precision, we are currently running the experiments and will add the results in the following days.
> > >
> > > - I understand, but maybe there would be a less ambiguous word for it? e.g. Independent sampling.
> > >
> > > This would be indeed a less ambiguous word, we will modify the wording when adding the new experiments results with the Spherical Sliced Wasserstein sampling.

---

> > > ### Author Response · Authors · 2025-04-10
> > > **Modification for reviewer 1**
> > >
> > > Thank you again for your comments.
> > >
> > > We added a section dedicated to the method proposed in [2] and the resulting experimental comparisons.
> > >
> > > We also replaced the wording "incremental computation" to "independent computation".

---

> > > > ### Comment · Reviewer_b6ih · 2025-04-13
> > > >
> > > > Thank you for the update and the additional experimental comparisons.
> > > >
> > > > A last quick remark on the Section 3.3 with the description of the method based on the Spherical-Sliced Wasserstein distance. It is stated that it is "pseudo-distance". Actually, it has been shown that it is a distance at least in some cases (see Theorem 5.1 in [1] on $S^2$ and Theorem D.3 in [2] in any dimension but between absolutely continuous measures).
> > > >
> > > > [1] Quellmalz, M., Beinert, R., & Steidl, G. (2023). Sliced optimal transport on the sphere. Inverse Problems, 39(10), 105005.
> > > >
> > > > [2] Liu, X., Bai, Y., Martín, R. D., Shi, K., Shahbazi, A., Landman, B. A., ... & Kolouri, S. (2024). Linear Spherical Sliced Optimal Transport: A Fast Metric for Comparing Spherical Data. arXiv preprint arXiv:2411.06055.

---

> > > > > ### Author Response · Authors · 2025-04-14
> > > > > **Modification for reviewer 1**
> > > > >
> > > > > Thank you for the precision.
> > > > >
> > > > > We added a remark (Remark 5) accordingly in this specific section.

---

### Review · Reviewer_BLhD · 2025-03-14

**Summary Of Contributions:**

This paper conducts a comparative study of various sampling methods from the unit sphere for computing the sliced Wasserstein distance between two distributions. The authors examine three main classes of sampling strategies: (i) random sampling (uniform and orthonormal), (ii) discrepancy-based sampling (e.g., Riesz points, Sobol, Halton), and (iii) variance reduction techniques (e.g., spherical harmonics control variate).

The main contribution of the paper is the comprehensive comparison between these methods, providing practical guidelines on selecting the most suitable strategy based on the problem's dimensionality and computational constraints.

Additionally, the paper establishes the regularity of the (1D) Wasserstein distance between the projections of two discrete measures along a given direction $\theta$.

**Audience:**

Yes

**Claims And Evidence:**

Yes

**Requested Changes:**

- In section 2.2., it lacks discussions on how to prove proposition 1 when $\mu$ and $\nu$ are continuous.
- Right after Proposition 3, it is unclear to me if $Y_M$ is an unbiased/consistent estimator for the sliced Wasserstein distance. It said "As a consequence, this method inherits the properties of classical Monte Carlo, with a rate of convergence independent of the dimension of the input measures", referring to the variance inequality. However, it did not say anything about the bias, so it is not enough to conclude.
- Three methods in Section 3.2.1.4. would need proper references.
- In the experiments, it is unclear to me why some methods are missing from certain dimensions. For example, in Figure 4.1, the legends of the subplots are not at all consistent.
- The Koksma-Hlawka inequality is referenced a couple times in the discussion, yet no formal definition is provided.

Typo:

- In the caption of Figure 2.1 (b), $SW_2^2(_m u,\nu)$ should be $SW_2^2(\mu,\nu)$ .

**Strengths And Weaknesses:**

Strengths:

- The comparative study is thorough where most existing methods for sampling from a sphere are presented and experimented.
- The theoretical footings of each method are presented and discussed.

Weaknesses:

- This work is purely a comparative study and does not introduce any technical contributions.
- The insights are largely empirical.

---

> ### Author Response · Authors · 2025-03-30
> **Changes done on submission for reviewer 2**
>
> We want to thank the reviewer for their feedback. It allowed us to further improve our work.
>
> - In section 2.2, it lacks discussions on how to prove proposition 1 when $\mu$ and $\nu$ are continuous.
>
> Thank you for this suggestion, we have now added a proof that the function is Lipschitz (hence continuous and differentiable almost everywhere) for all probability measures $\mu$ and $\nu$, either continuous or discrete (see Proposition 1 page 5).
>
> - Right after Proposition 3, it is unclear to me if $Y_M$ is an unbiased/consistent estimator for the sliced Wasserstein distance. It said “As a consequence, this method inherits the properties of classical Monte Carlo, with a rate of convergence independent of the dimension of the input measures”, referring to the variance inequality. However, it did not say anything about the bias, so it is not enough to conclude.
>
> Indeed, we found an error in our manuscript in this paragraph and have corrected this part. $Y_M$ is unbiased because each angle $\theta$ of the orthonormal basis remains uniformly distributed, but nothing can be said about its consistency. We can not say anything about its MSE compared to the classic monte carlo estimator (see section 3.1.2 page 8).
>
> - Three methods in Section 3.2.1.4 would need proper references.
>
> We have added proper references in paragraph 3.2.1.4.
>
> - In the experiments, it is unclear to me why some methods are missing from certain dimensions. For example, in Figure, 4.1, the legends of the subplots are not at all consistent.
>
> Some methods are only defined in specific dimensions. For example all the methods with the name “Area Mapped” are only defined on $\mathbb{S}^2$, the Fibonacci sampling is also only defined on $\mathbb{S}^2$. In dimension 50, the S.H.C.V. method cannot be used due to the implementation limitations induced by the lack of knowledge on spherical harmonics on higher dimensions.
> We have also added a table giving details on which dimensions we used for each method, see Table 3 p.17.
>
> - The Koksma-Hlawka inequality is referenced a couple times in the discussion, yet no formal definition is provided
>
> The Koksma-Hlawka inequality was formally defined in Proposition 4, we have added the name to the proposition for more clarity.

---

### Review · Reviewer_u77K · 2025-03-17

**Summary Of Contributions:**

The paper provides a comprehensive survey of various sampling strategies used in the estimation of the Sliced Wasserstein distance. It revisits key theoretical properties of the SW distance—such as its continuity, piecewise smoothness, and Lipschitz behavior—and uses these properties to motivate a taxonomy of sampling methods. In particular, the authors detail three broad classes of approaches: random samplings, discrepancy-based approaches, and variance reduction techniques. Some numerical experiments are shown to accompany the methodologies presented.

**Audience:**

Yes

**Broader Impact Concerns:**

Not applicable.

**Claims And Evidence:**

Yes

**Requested Changes:**

- The authors should add more intuitive explanations alongside the technical derivations—especially regarding the behavior of the discrepancy measures and the curse of dimensionality—to make the material more accessible.
- It would also be beneficial to include a more in-depth discussion of the limitations observed in the experiments, such as scalability issues and the effect of hyperparameter choices on convergence.
- The authors should provide a more detailed description of the experimental setup (e.g., parameter settings and implementation choices) or direct readers to a well-documented repository to ensure reproducibility.
- A dedicated section discussing the trade-offs between theoretical guarantees and actual computational overhead in practice.

**Strengths And Weaknesses:**

Strengths:

- The work is very thorough, covering both theoretical aspects and practical implementations.
- A clear view of sampling methods is provided. This will help readers understand trade-offs (e.g., between convergence rate and computational overhead).
- The experimental section is broad, spanning synthetic Gaussian mixtures, persistence diagrams, 3D point clouds, and MNIST.

Weaknesses:
- Some parts of the paper are quite dense and technical, which may pose challenges for practitioners who are less familiar with advanced numerical integration and optimal transport theory.
- Some sections (especially the theoretical derivations) could benefit from additional intuition or summary to guide readers through the complex analysis.
- The experimental comparisons, while broad, could be more detailed in discussing the limitations or failure cases of some methods, especially in very high-dimensional settings.
- Implementation details (e.g., parameter choices for optimization routines or sensitivity to hyperparameters) are mentioned but might require deeper discussion for reproducibility.

---

> ### Author Response · Authors · 2025-03-30
> **Changes done on submission for reviewer 3**
>
> We want to thank the reviewer for their feedback.
>
> - The authors should add more intuitive explanations alongside the technical derivations - especially regarding the behavior of the discrepancy measures and the curse of dimensionality - to make the material more accessible.
>
> Thank you for your feedback. We have added an intuitive explanation regarding the behavior of the discrepancy measures of low discrepancy sequences and the curse of dimensionality on the unit d-cube in remark 2 page 11.
>
> - It would also be beneficial to include a more in-depth discussion of the limitations observed in the experiments, such as scalability issues and the effect of hyperparameter choices on the convergence
>
> In terms of scalability, the s-Riesz point configuration does not fair well with a very large sampling number. Typically, one would need 9x10^7 seconds to generate 10^6 points. We added a remark regarding this point (see remark 8 p 18)
>
> - The authors should provide a more detailed description of the experimental setup (e.g, parameter settings and implementation choices) or direct readers to a well-documented repository to ensure reproducibility.
>
> We have added more specifications about the libraries used in our experiments, and where to find the code used for generating the s-Riesz points in section 4.1. We also modified the anonymous repository to ensure the reproducibility of figure 4.1 at least.
>
> - A dedicated section discussing the trade-offs between theoretical guarantees and actual computational overhead in practice.
>
> We have completed the recommendation section (section 5) with specific comments on these trade-offs. We have also added a table giving the time complexity and space complexity of each method (table 2 p.7).

---

### Decision · Action_Editor_mPku · 2025-05-14

**Recommendation:** Accept as is

**Comment:**

This paper presents a comprehensive survey and comparative analysis of sampling strategies for approximating the Sliced Wasserstein (SW) distance. It begins with a theoretical review of the SW distance’s continuity, smoothness, and Lipschitz properties, which motivate a taxonomy of estimation methods. The authors categorize sampling strategies into three main classes: (i) random sampling (uniform, orthonormal), (ii) discrepancy-based methods (e.g., Sobol, Halton, Riesz), and (iii) variance reduction techniques (e.g., spherical harmonics control variate). The paper includes complexity analysis, theoretical insights, and practical guidelines based on numerical comparisons across different settings and dimensionalities.

Overall, this paper provides a valuable summary of important sampling techniques for the Sliced Wasserstein distance and is a meaningful contribution to the machine learning community. All reviewers agree on accepting the paper, and their concerns have been adequately addressed. Therefore, I also recommend acceptance.

**Audience:**

Optimal transport is a popular topic of machine learning. Thus, TMLR's audience will be interested in the work.

**Claims And Evidence:**

The paper's claim is well supported by theorems and experiments.